# In-situ adsorption-coupled-oxidation enabled mercury vapor capture over *sp*-hybridized graphdiyne

Honghu Li [1,2,5], Chuanqi Pan[1,3,5], Xiyan Peng[1,2], Biluan Zhang[1,3], Siyi Song[1,3], Ze Xu[1,3], Xiaofeng Qiu[1,3], Yongqi Liu[1,3], Jinlong Wang[1,3,4] & Yanbing Guo [1,3,4] ✉

Developing efficient and sustainable carbon sorbent for mercury vapor (Hg[0]) capture is significant to public health and ecosystem protection. Here we show a carbon material, namely graphdiyne with accessible *sp*-hybridized carbons (HsGDY), that can serve as an effective "trap" to anchor Hg atoms by strong electron-metal-support interaction, leading to the in-situ adsorption-coupled-oxidation of Hg. The adsorption process is benefited from the large hexagonal pore structure of HsGDY. The oxidation process is driven by the surface charge heterogeneity of HsGDY which can itself induce the adsorbed Hg atoms to lose electrons and present a partially oxidized state. Its good adaptability and excellent regeneration performance greatly broaden the applicability of HsGDY in diverse scenarios such as flue gas treatment and mercury-related personal protection. Our work demonstrates a *sp*-hybridized carbon material for mercury vapor capture which could contribute to sustainability of mercury pollution industries and provide guide for functional carbon material design.

The high-volatile and water-insoluble gaseous elemental mercury (Hg[0]) is difficult to curb and can migrate long distances by the atmosphere[1–5], becoming a globally dangerous pollutant. High concentrations of mercury have been detected in fish, agricultural crops, animals over the world and even in the polar regions[6,7]. Deposited Hg[0] can be biotically converted into methyl mercury (MeHg) with strong lipophilicity[8–10], which is prone to bioaccumulate in the food chain and eventually brings serious harm to human health by causing nervous and cardiovascular diseases, cancer, and even sudden death. Industrial activities such as coal-fired power generation, non-ferrous metal smelting, cement production and measurement/electronic instruments manufacturing greatly contribute to the release of mercury into environments and have raised total atmospheric mercury concentrations by about 450% above natural levels[11–14]. Considering the serious harmful effects of mercury, developing efficient and sustainable mercury vapor removal technologies is mandatory and has attracted a great deal of interest recently.

Standing out from various mercury abatement methods[15–27], adsorption holds great promise due to its low cost, simplicity and effectiveness. The porous structure and environmental friendliness of carbon material makes it a promising candidate for mercury vapor capture. Unfortunately, the weak interfacial bonding with mercury is still the choke point in efficient mercury capture by carbon material. Although introducing ligands with affinity towards mercury such as halogen, sulfur and selenium species can enhance mercury adsorption[28–31], the possible leakage and loss of introduced species (e.g., toxic selenium) during sorbent application and regeneration would cause the performance decline as well as secondary pollution. The aforementioned problems necessitate the search for more efficient, sustainable and environmental-friendly carbon materials for mercury capture. As is known, metal ions (e.g., Hg[2+] and Pb[2+]) can be immobilized on alkynyl groups via soft acid-soft base interactions[32,33]. Such complexation effect is also the foundation of catalytical

[1]Institute of Environmental and Applied Chemistry, College of Chemistry, Central China Normal University, Wuhan, PR China. [2]Research Center for Environment and Health, School of Information Engineering, Zhongnan University of Economics and Law, Wuhan, PR China. [3]Engineering Research Center of Photoenergy Utilization for Pollution Control and Carbon Reduction, Ministry of Education, College of Chemistry, Central China Normal University, Wuhan, PR China. [4]Wuhan Institute of Photochemistry and Technology, Wuhan, PR China. [5]These authors contributed equally: Honghu Li, Chuanqi Pan. ✉e-mail: guoyanbing@mail.ccnu.edu.cn

hydrochlorination of $C_2H_2$ by $HgCl_2$, which is widely applied in the synthesis of vinyl chloride monomer[34]. Inspired by this chemical nature of $sp$-hybridized carbon (C≡C), we hypothesized that acetylenic bond-rich carbon materials might have certain adsorption capacity for mercury vapor. However, different from directly adsorption of positively charged $Hg^{2+}$ via electrostatic attraction and covalent bonding, the adsorption of electrically neutral $Hg^0$ on carbon surface is more challenging. It requires an in-situ $Hg^0$ adsorption-coupled-oxidation process, hardly achieved by traditional carbon materials such as $sp^2$-hybridized (C=C) graphene, which has not been demonstrated over carbon-based sorbents.

To tackle the aforementioned challenges, a carbon material featured by hexagonal pore structure (16.3 Å) and extended π-conjugated carbon skeleton composed of aromatic rings and acetylene linkages, namely hydrogen substituted graphdiyne (HsGDY), was synthesized and utilized for mercury vapor capture. The rapid diffusion of $Hg^0$ and effective interfacial electron transfer to form strongly-bonded Hg greatly boosts the immobilization of gas-phase $Hg^0$ on HsGDY, which is benefited from the large hexagonal pore structure and uneven surface charge distribution of HsGDY. Such in-situ adsorption-coupled-oxidation over HsGDY enables efficient mercury vapor capture with adsorption capacity of 0.71 μg/m², ~23.66 and 17.75 times as much as graphdiyne (GDY) and activated carbon (AC), respectively. Furthermore, HsGDY can achieve excellent regeneration performance via the adsorption-desorption process accompanied by reversible electron transfer between Hg and HsGDY. More encouraging is its adaptability to the complex industrial gas conditions such as high $SO_2$ content and temperature fluctuation. We demonstrate that HsGDY has excellent $Hg^0$ capture ability and the in-situ adsorption-coupled-oxidation over such $sp$-hybridized carbon material may offer more outstanding answers and provide guide for $Hg^0$ capture as well as carbon material design, which attracts attention in future.

## Results

### The theoretically performance advantage of HsGDY

The traditional $sp^2$-hybridized carbon (C=C) materials such as graphene (GE) show the common characteristics of uniform surface charge distribution, which might have weak charge transfer interaction with mercury atom. Especially, the newly emerged GDY is a two-dimensional planar periodic carbon allotropy formed by the direct connection of 1,3-diyne bonds ($sp$-hybridized carbon) and benzene rings ($sp^2$-hybridized carbon), which has a triangular hole structure composed of 18-C atoms with the inscribed circle diameter of about 5.46 Å as shown in Supplementary Fig. 1a. GDY with non-uniform surface charge distribution has superior charge transfer properties[35–44], possessing the potential for $Hg^0$ adsorption. Nevertheless, the atomic diameter of the mercury atom (3.42 Å) is close to the pore size of GDY, which might impact the mercury mass transfer and accessibility of mercury binding sites. HsGDY is a carbon-rich polymer, with unit composed of 42-C hexagons by connecting six benzene rings through butadiyne linkages (−C≡C−C≡C−)[45–47] as shown in Fig. 1a. Different from GE and GDY, HsGDY has lower atom density, larger pores (16.3 Å) and more $sp$-hybridized carbons in the pores, which might lead to higher $Hg^0$ adsorption capacity. Density functional theory (DFT) calculations were employed to reveal the adsorption sites and evaluate the effectiveness of HsGDY for $Hg^0$ capture. The atomic models and configurations of different typical carbon materials such as HsGDY, GDY, GE, and carbon nanotube (CNT) are displayed in Supplementary Fig. 2. The possible adsorption configurations and adsorption energies of the Hg atom on HsGDY, GDY, GE and CNT at different sites are displayed in Supplementary Tables 2−5. It can be seen that the calculated adsorption energies of $Hg^0$ onto HsGDY, GDY, GE, and CNT for the optimal adsorption configuration are 0.005 eV, −0.082 eV, 0.102 eV, and −0.398 eV, respectively. The calculated $Hg^0$ adsorption energy is somewhat related to but not perfectly consistent with the

$Hg^0$ adsorption performance. Adsorption is a complex process that involves many factors such as pore characteristics and surface chemical properties beyond just the calculated adsorption energy. For HsGDY, the Hg atom tends to be adsorbed at the side sites of acetylenic bond in plane with the adsorption energy of 0.005 eV. Overall, the adsorption energy of $Hg^0$ onto HsGDY is not negative but close to 0, indicating the relative stability and potential for $Hg^0$ adsorption over HsGDY with energy input.

The two-dimensional charge distribution of HsGDY and GDY based electron localization function (ELF) results are displayed in Fig. 1b and Supplementary Fig. 1b, respectively. It is implied that the charge distribution of HsGDY is more uneven compared to GDY. We quantitively compare the charge non-uniformity of GDY and HsGDY as shown in Supplementary Fig. 3. It can be seen that the charge non-uniformity (σ) of HsGDY is obviously higher than that of GDY. This uneven charge distribution of HsGDY might induce electron redistribution between the HsGDY and adsorbed Hg atom, leading to a strong interaction between them. The uneven charge distribution characteristics of HsGDY might be attributed to that the large pore structure leads to lower local atomic density and resultantly uneven distribution of $sp$-hybridized and $sp^2$-hybridized carbons in the local space. Additionally, in a single pore, HsGDY with hexagonal pore structure has 12 $sp$-hybridized carbons which are double that of GDY with triangular hole structure. We calculated the adsorption of multiple Hg atoms by HsGDY and GDY composed of 72 carbon atoms (the same quality), respectively. As shown in Fig. 1c and Supplementary Fig. 1c, HsGDY with large hexagonal pore structure can ultimately adsorb 6 Hg atoms through the in-plane acetylene bond side mode in one single unit space, while GDY only adsorbs 2 Hg atoms. This implies that the unique architecture of HsGDY improves the accessibility of $sp$-hybridized carbons that serve as binding sites for mercury.

Furthermore, we simulated the adsorption of Hg atom over multi-layer HsGDY and GDY based on DFT calculation since the ideal single-layer material is normally scarce as a result of π-π interactions and Van der Waals forces. As shown in Fig. 1d and Supplementary Fig. 1d, HsGDY tends to exist stably in an "AB" stacking configuration with an interlayer distance of 0.420 nm (2HsGDY). While the GDY layers are stacked through Van der Waals forces and π-π interactions, forming the most stable "ABC" stacking structure with an interlayer distance of 0.365 nm (3GDY). For 2HsGDY, the Hg atom also tends to be adsorbed at the side sites of acetylenic bond in plane with the adsorption energy of −0.008 eV (Supplementary Table 6). It is relatively difficult for Hg atom to be adsorbed in the interlayer of 2HsGDY. This indicates that the rapid diffusion of Hg atoms can be achieved between layers of HsGDY (Fig. 1e). While for 3GDY, the Hg atom tends to be adsorbed at the center of triangular hole of GDY as well as the interlayer of 3GDY (Supplementary Table 7). The diffusion of Hg atoms between layers of 3GDY should overcome the effect of interlayer adsorption. In addition, the pore diameter of GDY (5.46 Å) is much smaller than HsGDY (16.3 Å). The staggered "ABC" stacking structure of 3GDY would result in a tortuous diffusion path of Hg atoms as shown in Supplementary Fig. 1e. Hence, the diffusion resistance of Hg atoms in 3GDY is larger compared to 2HsGDY, which is not conducive for the rapid diffusion of Hg atoms between carbon layers. The calculations of the adsorption of multiple Hg atoms by 2HsGDY and 3GDY indicate that 2HsGDY can adsorb 16 Hg atoms (Fig. 1f) while 3GDY only adsorbs 4 Hg atoms (Supplementary Fig. 1f) under the same quality conditions. The aforementioned analysis indicates that HsGDY theoretically has better $Hg^0$ adsorption performance than other carbon materials.

The physico-chemical structure of as-prepared HsGDY was investigated. The SEM image results show that HsGDY is formed by the interconnection of nanospheres (Supplementary Fig. 4a). The interlayer distance of HsGDY is about 0.416 nm, which is the characteristic value for the multilayer HsGDY (Fig. 1g)[45]. The Raman spectra for HsGDY as shown in Fig. 1h exhibit four prominent peaks at 1358 cm⁻¹,

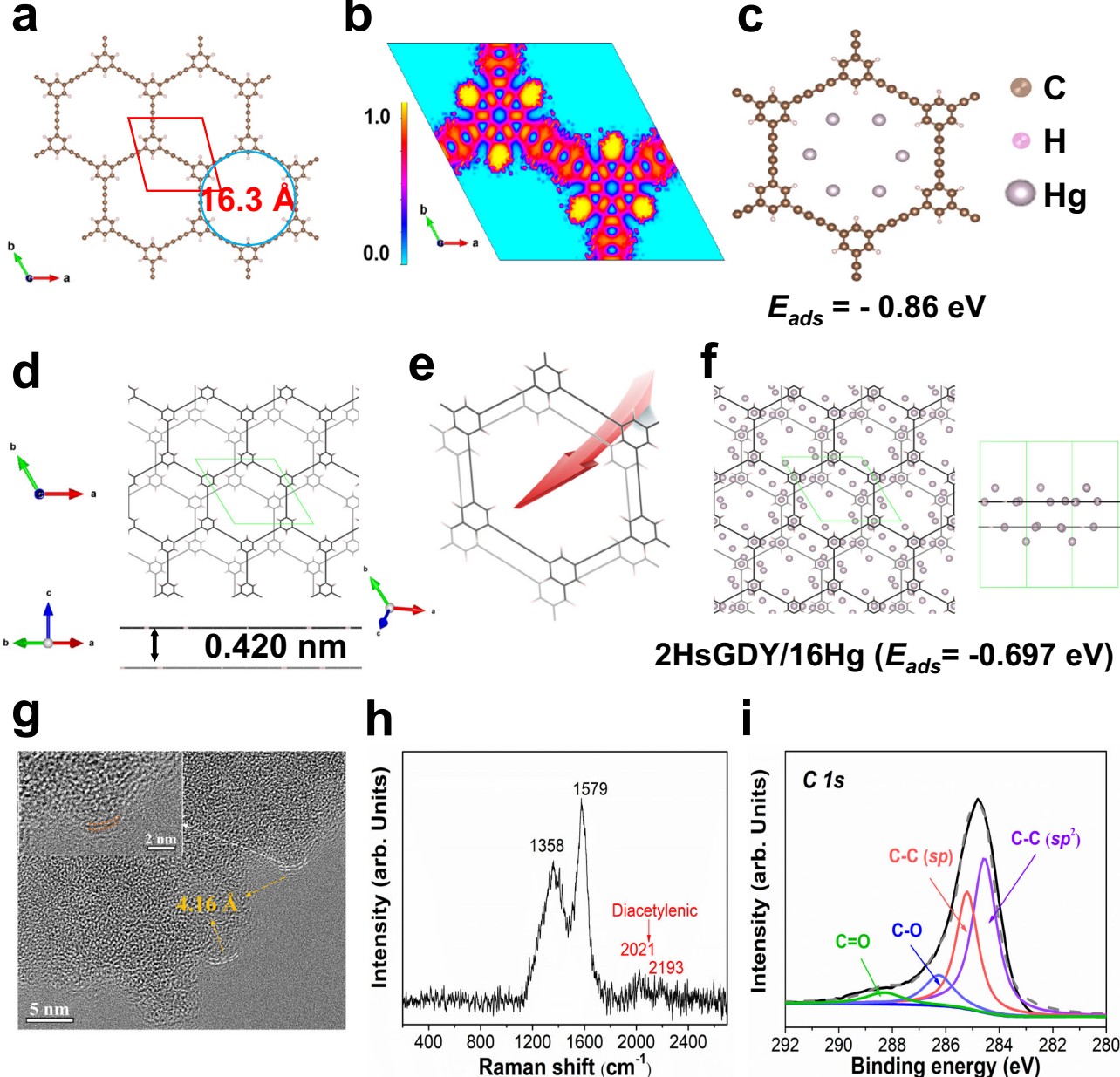

**Fig. 1 | Physico-chemical structure of HsGDY and its theoretical performance.**
**a** The molecular structure of HsGDY; **b** The 2D charge distribution of HsGDY (The value 0 to 1 implies the probability of electron localization); **c** HsGDY adsorbs multiple mercury atoms at the same quality (72 carbon atoms); **d** The atomic model and configuration of 2HsGDY; **e** The diffusion path of Hg atoms in 2HsGDY; **f** 2HsGDY adsorbs multiple mercury atoms (left figure for top view and right figure for side view); **g** HRETM image of fresh HsGDY (inset for the magnified image); **h** Raman spectra of fresh HsGDY; **i** *C1s* XPS spectra of fresh HsGDY.

1579 cm⁻¹, 2021 cm⁻¹, and 2193 cm⁻¹ due to its rich aromatic rings and acetylenic bonds[47]. The *C1s* peaks of HsGDY (Fig. 1i) can be deconvoluted into four subpeaks of C-C ($sp^2$) at 284.6 eV, C-C ($sp$) at 285.2 eV, C-O at 286.5 eV and C=O at 288.3 eV, respectively[47]. The ~1:1 area ratio of C-C ($sp^2$) to C-C ($sp$) is consistent with the typical carbon skeleton of HsGDY. The pore structure of HsGDY was further studied by nitrogen adsorption-desorption experiments. It can be seen from Supplementary Fig. 4b that the adsorption quantity of $N_2$ is sharply increased at very low $P/P_O$ due to a mass of single-layer adsorption of $N_2$ in the micropores. The specific surface area (SSA) calculated by Brunauer–Emmett–Teller (BET) method for the sample is 160.3 m² g⁻¹. The pore size distribution of HsGDY is centered at about 0.8 nm (Supplementary Fig. 4c), indicating a highly ordered porous structure in HsGDY. To summarize, the unique architecture, acetylenic bond

structure, and uneven charge distribution characteristics of HsGDY might endow it with potential $Hg^0$ capture performance.

**The practical applicability of HsGDY**
The applicability of HsGDY for gas-phase $Hg^0$ capture was examined under $N_2$ flow containing 340 μg m⁻³ $Hg^0$ by a laboratory-scale fixed-bed reaction system (Supplementary Fig. 5 and Supplementary Table 1). The performances of GDY, CNT, GE, and AC were also explored for comparative analysis. As displayed in Fig. 2a, the $Hg^0$ adsorption efficiency of HsGDY exceeds 90%, which is apparently superior to that of GDY, CNT, GE, and AC. The adsorption capacity during 90 min was determined as shown in Fig. 2b. HsGDY shows the highest adsorption capacity of 113.5 μg g⁻¹ among the tested samples, which is ~17 times as much as GDY. The specific surface areas of GDY,

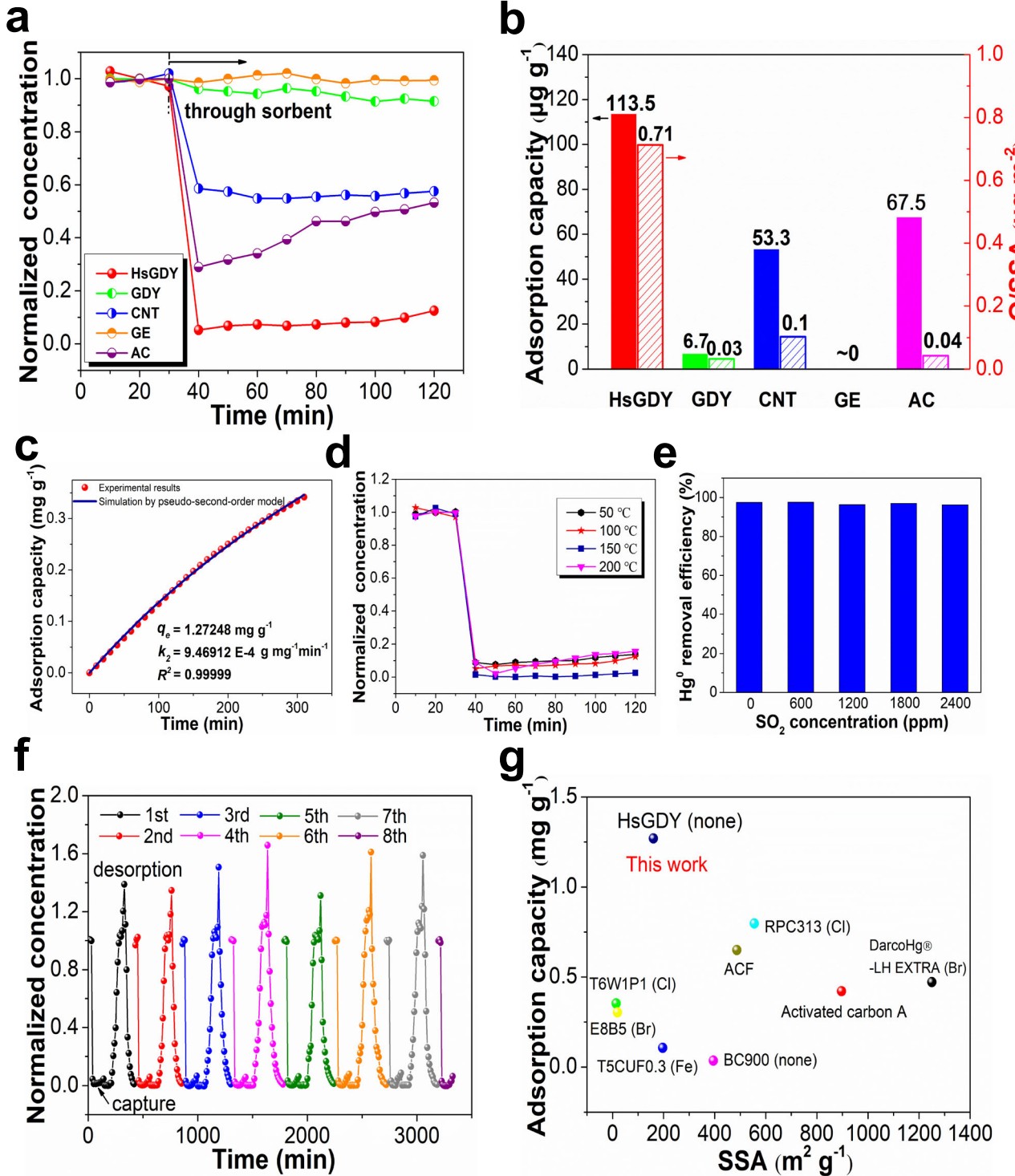

**Fig. 2 | The Hg⁰ adsorption performance evaluation. a** The Hg⁰ adsorption performances of different carbon sorbents (dash line indicates time point for gas flow passing through sorbent); **b** The adsorption capacities and Q/SSA of different carbon sorbents; **c** The experimental data analyzed by pseudo-second order model; **d** The Hg⁰ adsorption performance of HsGDY in the temperature range of 50–200 °C; **e** The effect of $SO_2$ on Hg⁰ adsorption over HsGDY; **f** The performance of HsGDY during 8 adsorption-desorption cycles; **g** Comparison of HsGDY with other reported carbon sorbents (modification elements in the brackets) (see Supplementary Table 8 for the details).

CNT, GE, and AC were determined to be 210.7, 527.9, 38.8, and 1583.3 $m^2\,g^{-1}$, respectively. The variation of Hg⁰ adsorption capacity with average pore volume and pore width is displayed in Supplementary Fig. 6. As shown in Fig. 2b, HsGDY also exhibits the highest value of Q/SSA (adsorption capacity divided by SSA), which is significantly higher than that of other samples. Above results imply that

the pore structure of the material is not a decisive factor for Hg⁰ adsorption and the specific mechanism must be existed for the Hg⁰ capture over HsGDY.

The 50%-breakthrough curve of HsGDY for Hg⁰ capture is shown in Supplementary Fig. 7 and the corresponding adsorption capacity curve is displayed in Fig. 2c. In order to elucidate the interaction

between $Hg^0$ and HsGDY, the experimental data was analyzed by pseudo-first-order model (Supplementary Fig. 8a), pseudo-second-order model (Fig. 2c) and Weber-Morris model (Supplementary Fig. 8b), respectively. As can be seen, the correlation coefficient ($R^2$) for the pseudo-second-order model describing the chemical adsorption nature is 0.99999, which is higher than that for the pseudo-first-order model (external mass transfer) and Weber-Morris model (intra-particle diffusion)[48]. This result implies that the adsorption of $Hg^0$ onto HsGDY is mainly a chemisorption process. The formed mercury compounds on HsGDY surface were further examined by the temperature-programmed desorption of Hg (Hg-TPD) analysis. As shown in Supplementary Fig. 9a mercury desorption peak emerges at around 320 °C for HsGDY which can be ascribed to the decomposition/desorption of strongly bonded mercury species, e.g., oxidized mercury[49]. This result demonstrates an interaction between $Hg^0$ and HsGDY with the occurrence of charge transfer. While for GDY, a mercury desorption peak at 160 °C is observed, which can be attributed to the weakly bonded mercury species. Moreover, the adsorption capacity of HsGDY at equilibrium is determined to be 1.27 mg g$^{-1}$, which is superior to GDY and most of the other reported carbon sorbents such as the commercial Br-modified carbon sorbent DARCO® Hg-LH EXTRA (0.47 mg g$^{-1}$)[50] (Fig. 2g and Supplementary Table 8).

Furthermore, HsGDY might have broad applicability for $Hg^0$ capture due to its excellent $SO_2$ resistance and regeneration property. As shown in Fig. 2d, the $Hg^0$ adsorption efficiency of HsGDY is stable over around 90% in the temperature range of 50–200 °C. The slight decrease of adsorption efficiency at 200 °C is attributed to that the $Hg^0$ desorption will prevail over the adsorption at high temperature, which is verified by the Hg-TPD result as shown in Supplementary Fig. 9. Additionally, we adopted simple thermal treatment to regenerate the Hg-laden HsGDY. As shown in Fig. 2f, the $Hg^0$ adsorption efficiency does not decline significantly in the 8 cycles of $Hg^0$ capture and regeneration (thermal treatment at 400 °C under $N_2$ stream), which facilitates its reusability and cost reduction. Considering that $SO_2$, NO, and $H_2O$ are typical components in real flue gas and might influence $Hg^0$ removal, we further examined the $Hg^0$ adsorption performance of HsGDY in the gas stream containing different contents of $SO_2/NO/H_2O$. As shown in Supplementary Fig. 10, NO slightly promotes (at least not inhibitory) the $Hg^0$ adsorption over HsGDY. $H_2O$ presence slightly decreases the $Hg^0$ removal efficiency (from 97.4% to 87.7% under 3% vol. $H_2O$) over HsGDY, which can be attributed to the competitive adsorption between $Hg^0$ and $H_2O$ over the adsorption sites. As shown in Fig. 2e, the $Hg^0$ removal efficiency of HsGDY maintains at around 97.5% under 600–2400 ppm $SO_2$ conditions. As shown in Supplementary Fig. 11, the adsorption energies of the $SO_2$ molecule at different adsorption sites over HsGDY are all nearly 3.5 eV, which is obviously higher than that of Hg atom (-0 eV). Above exciting results manifest that $Hg^0$ is preferentially absorbed on HsGDY, leading to its excellent anti-$SO_2$ ability.

From the viewpoint of real application, the $Hg^0$ capture by HsGDY can be coupled to the dust removal system currently equipped in coal-fired power plants. Due to the excellent $Hg^0$ capture performance of HsGDY, we proposed an improved bag filter system to efficiently capture $Hg^0$ from flue gas. As shown in Supplementary Fig. 12a, HsGDY can be injected into the flue gas duct upstream the separate bag filter after dust removal. $Hg^0$ in the gas stream will be immobilized on the HsGDY filter cake formed on the filter medium. The Hg-laden HsGDY can be recycled by dust cleaning such as mechanic vibration and then regenerated by thermal treatment. Specially, the aforementioned filter unit for $Hg^0$ capture can be coupled into an integrated bag filter by partition design. As shown in Supplementary Fig. 12b, the filter medium can be modified by HsGDY and then immobilizes $Hg^0$ from flue gas. The fly ash can be removed from the filter medium by reverse airflow (hot) cleaning. In the meantime, the desorption of mercury species will occur and the concentrated mercury is recovered by

condensation which can be applied to industrial production. Such technical route requires the good high-temperature resistance performance of the filter medium.

We selected two typical commercial filter mediums i.e., glass-chemical compound fibers filter material (denoted as FMS) and polytetrafluoroethylene (PTFE), and examined their $Hg^0$ adsorption performances without and with HsGDY modification. Briefly, HsGDY was first dispersed in deionized water by ultrasonic treatment to form a suspension. Then HsGDY was attached onto the filter medium via a vacuum filtration method. The macroscopic images and SEM images of the filter mediums are displayed in Fig. 3a and Fig. 3b, respectively. It can be seen from Fig. 3b that the HsGDY nano-particles with the layer thickness of ~ 115 nm are successfully coated on the surface of FMS fibers. The pictures of the different filter mediums placed in the quartz tubes are shown in Fig. 3c. The simulated flue gas containing 340 μg m$^{-3}$, 600 ppm $SO_2$, 600 ppm NO, 5% vol. $O_2$, 3% vol. $H_2O$ and balanced $N_2$ passed through the filter mediums and the outlet $Hg^0$ concentration was detected. The $Hg^0$ adsorption performances of different filter mediums are displayed in Fig. 3d. It can be seen that non-modified FMS and PTFE are not capable of $Hg^0$ capture. However, HsGDY-FMS and HsGDY-PTFE show excellent $Hg^0$ removal abilities. ~96% and ~91% of $Hg^0$ can be respectively removed by HsGDY-FMS and HsGDY-PTFE. This result indicates that the $Hg^0$ removal performance of filter medium can be greatly enhanced when HsGDY is attached to the filter medium, envisioning the possibility that the $Hg^0$ capture by HsGDY can be coupled to the dust removal system currently available.

Additionally, mercury is generally utilized in the manufacturing of scientific measurement instruments (such as barometers, thermometers, etc.) and electronic products. Extensive and long-term exposure to mercury in the production workshop will result in acute/chronic mercury poisoning and cause diseases of nerve, respiration, digestion, and urinary systems. There remains high concern for the populations with occupational exposure to mercury. Strengthening indoor ventilation and personal protection is of great significance to meet the occupational health requirements. Based on the above studies, HsGDY can be applied to the personal protective equipment (PPE) and ventilation system to prevent the workers from mercury exposure and remove the mercury vapor in the exhaust gas, as schematically illustrated in Fig. 3e. Superior to sulfur and selenium compounds with high mercury uptake capacity while high toxicity[51], environmental-friendly pure carbon material, namely HsGDY, can be directly utilized in the filter layer of mask. HsGDY can be also attached to the filter medium in ventilation duct to remove mercury vapor from the gas stream. In addition, the mercury can be desorbed from the mercury-laden HsGDY by acid washing. The leached mercury in the solution can be effectively precipitated by $Na_2S$ and the formed HgS (small particles as shown in Supplementary Fig. 13) can be recovered and processed into raw materials for industrial production again. Hence, HsGDY is a sustainable carbon material with ultrahigh efficient mercury vapor capture and promising application prospects.

## In-situ adsorption-coupled-oxidation mechanism

To better understand the ultrahigh efficient mercury vapor capture over HsGDY, the interaction mechanism between $Hg^0$ and HsGDY is elucidated in detail. As displayed in Supplementary Fig. 14a, the Raman spectra of HsGDY after $Hg^0$ adsorption (Hg/HsGDY) presents a shift in peak position and an increase in peak intensity compared to HsGDY, indicating the formation of coordination bonds between carbon atoms of conjugated diyne linkers (2021 cm$^{-1}$ and 2193 cm$^{-1}$)[52–54] in HsGDY and Hg atoms. The increase in the intensity of the acetylene bond peak may be attributed to the Raman enhancement effect of Hg. The *Hg 4f* XPS analysis of Hg/HsGDY as shown in Supplementary Fig. 14b further reveals that gas-phase Hg atoms are successfully adsorbed on the HsGDY surface, mainly present in the form of partially oxidized state[55]. The morphology structure of Hg/HsGDY as shown in HRTEM image

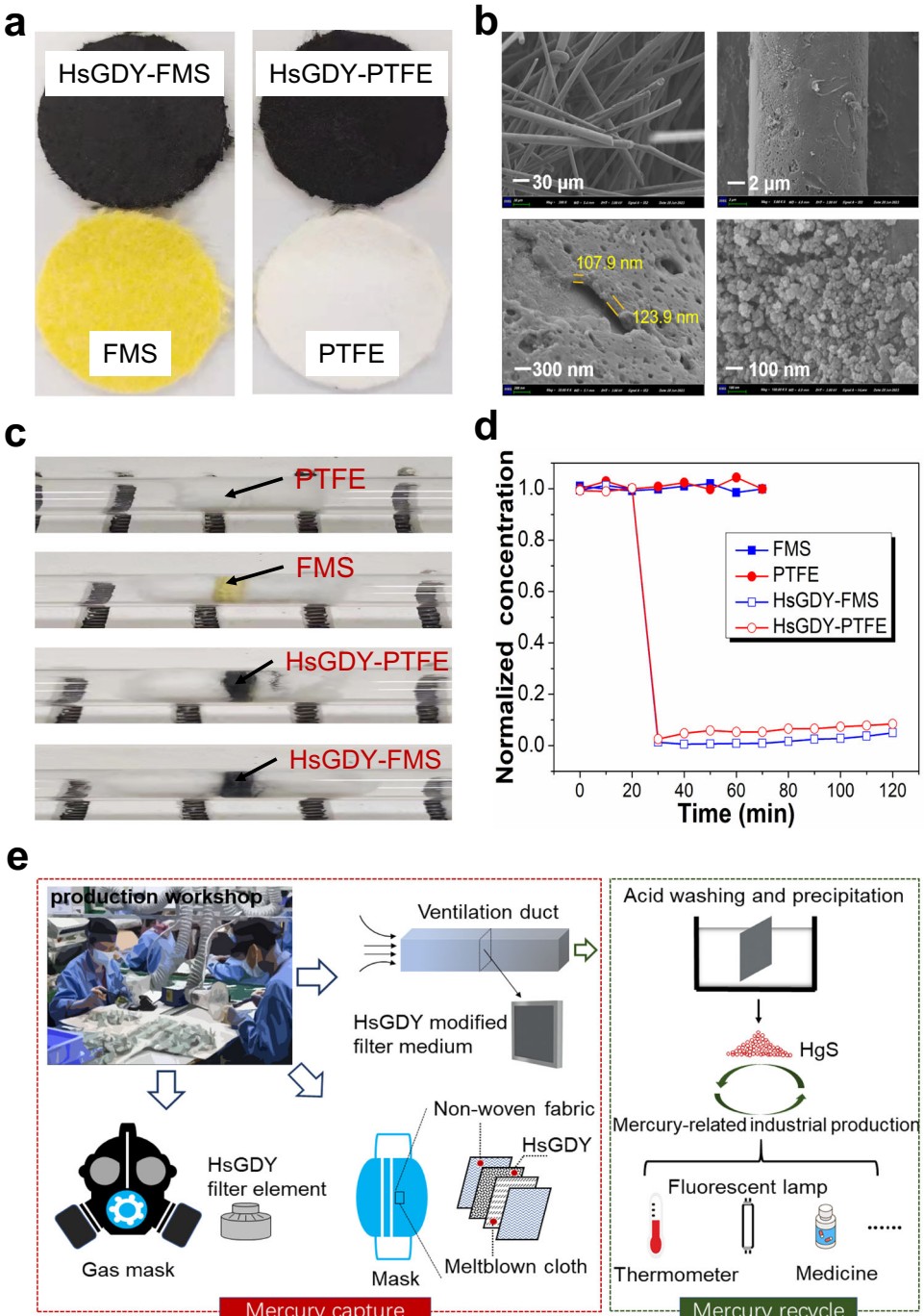

**Fig. 3 | The Hg⁰ capture by HsGDY modified filter mediums. a** The macroscopic images of HsGDY-FMS, HsGDY-PTFE, FMS, and PTFE; **b** The SEM images of HsGDY-FMS with different magnifications (the two sub-figures at top indicate the fiber structure of HsGDY-FMS; the sub-figure at the bottom left corner indicates the thickness of HsGDY coating layer labeled by 107.9 and 123.9 nm; the sub-figure at the bottom right corner indicates the particle structure of HsGDY over FMS fiber); **c** The pictures of the different filter mediums placed in the quartz tubes; **d** The Hg⁰ adsorption performances of different filter mediums; **e** The schematic diagram of mercury capture by HsGDY in the production workshop and mercury recycle.

(Fig. 4a) and the corresponding energy dispersive X-ray spectroscopy (Supplementary Fig. 15 and Supplementary Fig. 16) indicate that the Hg atoms adsorbed onto HsGDY surface exist in a highly dispersed state. It can be further confirmed by the HAADF-STEM results that the isolated Hg atoms (bright) are clearly observed on the surface of spent HsGDY (relatively dark) as shown in Fig. 4b (Supplementary Fig. 17 for fresh HsGDY).

DFT calculations were conducted to further reveal the Hg atom adsorption behavior and the binding mechanism over *sp*-hybridized carbon (HsGDY) compared to *sp²*-hybridized carbon (GE). The charge density difference in Fig. 4d indicates that significant electron redistribution occurs between *sp*-hybridized carbon and adsorbed Hg atom, in which electrons are accumulated on *sp*-hybridized carbon and depleted around the Hg atom. In detail, the Hg atom loses 0.27 e⁻ which is transferred to *sp*-hybridized carbon, thus existing in a partially oxidized state. This is consistent well with the XPS results of *Hg 4f*. The two-dimensional projection of differential charge density contours of Hg/HsGDY results also clearly illustrate this electron transfer behavior between Hg atom and *sp*-hybridized carbon (Fig. 4e). Additionally, the calculated partial density of states (PDOS) exhibits an orbital overlap

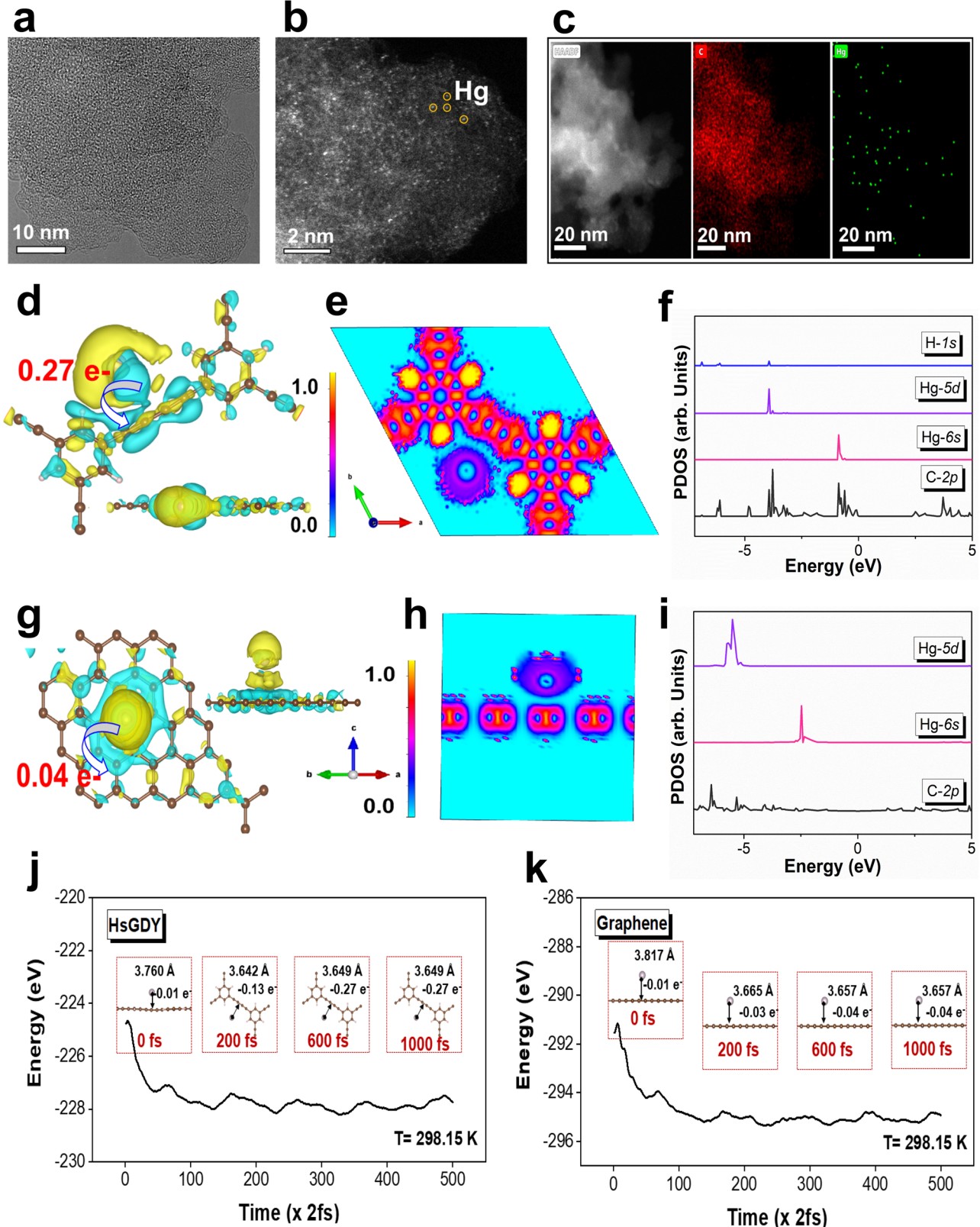

between *H 1s*, *Hg 5d*, *Hg 6s*, and *C 2p* below the Fermi energy level (~0 eV) for the adsorption configuration of Hg/HsGDY as shown in Fig. 4f. However, no obvious electron transfer and orbital overlap between Hg atom and *sp²*-hybridized carbon can be detected for the adsorption configuration of Hg/GE as shown in Fig. 4g, Fig. 4h and Fig. 4i, demonstrating the weak interfacial bonding between mercury and *sp²*-hybridized graphene. Moreover, the Ab initio

molecular dynamics (AIMD) simulation results as displayed in Fig. 4j, Fig. 4k, and Supplementary Fig. 18 clearly show the adsorption process of Hg atom from free state to stable adsorption state, accompanied with significant electron transfer especially for HsGDY. Additionally, the adsorption of Hg atom onto HsGDY can reach a relatively stable state with smaller energy fluctuations as shown in Supplementary Fig. 19.

**Fig. 4 | The Hg⁰ adsorption mechanism over HsGDY. a** The high-resolution transmission electron microscopy (HR-TEM) image of Hg/HsGDY; **b** The HAADF-STEM image of Hg/HsGDY (yellow circles for Hg atoms); **c** The HAADF-STEM image and corresponding element mapping images of C (red) and Hg (green); **d** The charge density difference of Hg/HsGDY (depletion and accumulation spaces are revealed in blue and yellow, respectively; the arrow represents the direction of electron transfer); **e** The two-dimensional projection of differential charge density contours of Hg/HsGDY; **f** The PDOS comparison for *Hg 5d, Hg 6s, C 2p,* and *H 1s* orbitals within Hg/HsGDY; **g** The charge density difference of Hg/GE (The arrow represents the direction of electron transfer); **h** The two-dimensional projection of differential charge density contours of Hg/GE; **i** The PDOS comparison for *Hg 5d, Hg 6s,* and *C 2p* orbitals within Hg/GE; **j** The molecular dynamic simulation results of Hg adsorption over HsGDY (inset figures for the adsorption configurations with time and the resultant electron transfer); **k** The molecular dynamic simulation results of Hg adsorption over GE (inset figures for the adsorption configurations with time and the resultant electron transfer).

Above results manifest that HsGDY with accessible *sp*-hybridized carbons can serve as an effective "trap" to anchor Hg atoms by strong electron-metal-support interaction, leading to the in-situ adsorption-coupled-oxidation of Hg. The adsorption process benefits from the unique architecture of HsGDY as previously discussed: the large hexagonal pore structure and "AB" stacking structure of HsGDY can facilitate the rapid diffusion of Hg atoms across the sorbent and make the mercury binding sites highly accessible. The oxidation process is driven by the surface charge heterogeneity of HsGDY which can itself induce the adsorbed Hg atoms to lose electrons and present a partially oxidized state. The favorable environment with convenient mass transfer channels and surface charge heterogeneity promotes mercury diffusion and electron exchange, thus enhancing the mercury binding ability of HsGDY. The Hg⁰ adsorption onto traditional carbon material is primarily due to the physical adsorption determined by the porous structure of carbon material. The interfacial bonding of Hg⁰ with the carbon surface is generally weak. The electron transfer between adsorbed Hg⁰ and carbon material can enhance the Hg⁰ immobilization on the carbon surface, which is hardly achieved by traditional *sp²*-hybridized graphene with uniform surface charge distribution. Although GDY is also a *sp*-hybridized carbon material similar to HsGDY, it is defeated by HsGDY for mercury capture due to its constrained mass transfer and relatively lower surface charge heterogeneity. Moreover, different from toxic sulfur and selenium compounds, the environmental-friendly HsGDY can be applied to personal protective equipment, greatly broadening its application fields. Overall, an in-situ Hg⁰ adsorption-coupled-oxidation process can be achieved by HsGDY, which enables its efficient mercury vapor capture and thus effectively solves the bottleneck problem of the weak interfacial bonding between mercury and traditional carbon sorbents.

Acting as the reverse process of adsorption, it can be expected that there might be a "reduction-coupled-desorption" process for Hg desorption. We first calculated the Hg atom desorption barrier of HsGDY through theoretical calculations. As shown in Fig. 5a, the desorption of Hg atoms requires overcoming certain energy barriers due to the electronic interaction between Hg and HsGDY and HsGDY can be restored to the original state after Hg desorption. This is consistent well with the XPS *Hg 4f* and Raman results (Fig. 5b and Fig. 5c). The characteristic peak of 104 eV ascribed to partially oxidized Hg is not detected in the regenerated HsGDY sample. Additionally, the Raman spectroscopy results show that the characteristic peaks in the regenerated HsGDY attributed to aromatic rings and acetylenic bonds are still retained and basically restored to the state of fresh sample. Since the mercury analyzer can only detect the elemental mercury, we employed Ontario Hydro Method (OHM) to identify the desorbed mercury species during thermal regeneration. The results displayed in Supplementary Fig. 20 show that the desorbed mercury species are primarily present in the form of elemental mercury. Hence, the adsorbed Hg atoms that have lost electrons to HsGDY recapture the electrons and are released to the gas stream in the form of elemental mercury by heat treatment, which can be further verified by the molecular dynamic simulation results of Hg desorption over HsGDY at 300 °C as shown in Fig. 5d. Therefore, HsGDY can achieve excellent regeneration performance via the adsorption-desorption process accompanied by reversible electron transfer between Hg and HsGDY as shown in Fig. 5e. Moreover, as shown in Supplementary

Fig. 21a and b, the regenerated HsGDY still exhibits a porous structure and its area ratio of C-C (*sp²*) to C-C (*sp*) is similar to that of fresh sample, indicating the structural stability of the carbon skeleton of HsGDY. The regeneration performance of sorbent is essential to its real application. Considering that the Hg⁰ adsorption efficiency of HsGDY does not decline significantly in the 8 cycles of Hg⁰ capture and regeneration, it is indicated that HsGDY has excellent regeneration property and application potential for mercury vapor capture.

## Discussion

Mercury poses serious hazards to the natural environment and public health due to its high toxicity, long atmospheric residence time, and bio-enrichment. Hence, developing efficient and sustainable mercury vapor removal technologies receives extensive attention. Carbon material sorbent exhibits high potential for mercury capture due to its porous structure and environmental friendliness. However, traditional carbon material cannot fulfill practical requirements for efficient elemental mercury capture due to its weak interfacial bonding with mercury. Owing to the large hexagonal pore structure, "AB" stacking structure and surface charge heterogeneity of HsGDY, HsGDY with accessible *sp*-hybridized carbons can realize an in-situ adsorption-coupled-oxidation of gas-phase elemental mercury, which enables its efficient mercury vapor capture and thus effectively solves the abovementioned bottleneck problem. The rapid diffusion of Hg⁰ and effective interfacial electron transfer to form adsorbed Hg with partially oxidized state greatly boosts the immobilization of gas-phase Hg⁰ on HsGDY, which has not previously been accessed in another carbon material. Furthermore, its adaptability to high SO₂ content and excellent regeneration performance greatly broadens the applicability of HsGDY in diverse scenarios such as flue gas treatment and mercury-related personal protection. In summary, the findings of this work demonstrate a carbon material for protecting human beings from the hazards of mercury pollution and contributing to the sustainability of mercury pollution industries in the post-Minamata Convention era.

## Methods

Synthesis of HsGDY. HsGDY powder was synthesized with a revised method through a classic in situ cross-coupling reaction of triethynylbenzene on copper foil as support in pyridine. In details, the cleaned copper foil was added to a three-necked flask along with 100 mL of pyridine. Subsequently, the 1,3,5-triethynylbenzene dissolved in 100 mL of pyridine was transferred to a constant-pressure dropping funnel and added dropwise to the three-necked flask at a rate that allowed the solution to be fully introduced within ~4 h. The reaction was then conducted at 100 °C under light shielding for 5 days. The obtained mixture was filtered under vacuum to obtain a yellow-brown powder. This powder was centrifuged with water, ethanol, and dimethyl sulfoxide (DMF) until the supernatant was colorless. After drying, the powder was refluxed in 2 M HCl at 80 °C for 6 h. The mixture was then washed with pure water until it reached a pH of 7. Subsequently, it was refluxed in 2 M sodium hydroxide solution at 80 °C for 6 h and washed with ultrapure water until it reached neutral again. The mixture was then centrifuged with water, ethanol, and DMF until the supernatant was colorless. The resultant product was dried under vacuum at 60 °C for 12 h. Under Ar atmosphere, the product was heated in a

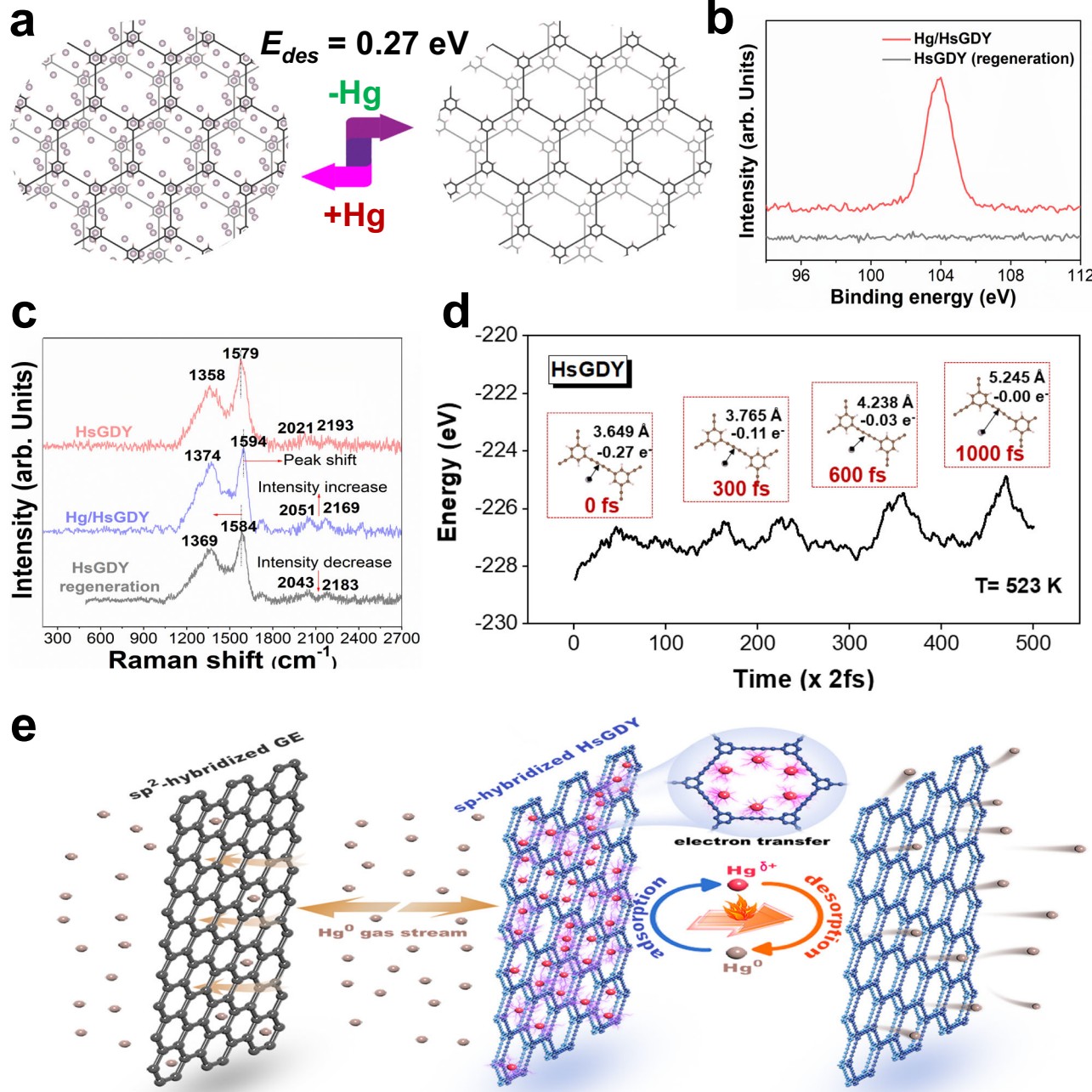

**Fig. 5 | The regeneration mechanism of HsGDY. a** The Hg desorption barrier of HsGDY; **b** The XPS *Hg 4f* spectrum of Hg/HsGDY and HsGDY after regeneration; **c** The Raman spectra of HsGDY, Hg/HsGDY and HsGDY after regeneration; **d** The molecular dynamic simulation results of Hg desorption over HsGDY (inset figures for the adsorption configurations with time and the resultant electron transfer); **e** The schematic diagram of $Hg^0$ adsorption-desorption process over *sp*-hybridized HsGDY ($Hg^{\delta+}$ denotes adsorbed Hg with partially oxidized state).

tubular furnace at a rate of 5 °C min$^{-1}$ to 400 °C and calcined for 2 h to obtain a tawny powder, which is HsGDY. All the chemicals (analytical grade) were purchased from Sinopharm Chemical Reagent Co., Ltd and used without further purification. Ultrapure water was used in all the experiments.

Structure characterization. High-resolution transmission electron microscopy (HR-TEM) images of the typical samples were obtained on a Tecnai G2 F30 apparatus at an accelerating voltage of 200 kV. High-angle annular dark-field scanning transmission electron microscopy (HAADF-STEM) images were obtained on the aberration-corrected cubed FEI Titan Cubed Themis G2 60–300 operated at 300 kV with cold filed-emission gun and double

hexapole Cs correctors (Thermo Fisher, USA). X-ray photoelectron spectroscopy (XPS) analysis was performed on a Thermo ESCALAB 250XI multifunctional imaging electron spectrometer, using monochromatic Al K radiation (1486.6 eV) operating at an accelerating power of 15 kW. The binding energy was calibrated using a *C 1s* peak at 284.8 eV as standard and quoted with an accuracy of ±0.1 eV. Raman spectroscopy were used to characterize molecular structure of the prepared materials. The Raman spectra were obtained by excitation at 532 nm using Raman Spectrometer (Lab-RAM HR JY-Evolution, 532 nm). Nitrogen adsorption/desorption measurements were recorded at 300 °C using a Micromeritics ASAP2020 gas-sorption system.

**Computational details.** All calculations using DFT in this work were carried out by Vienna ab initio simulation package (VASP 5.4.1)[56]. Exchange-correlation functions are taken as the generalized gradient approximation (GGA) in the form of Perdew–Burke–Ernzerhof (PBE)[57]. The projector augmented wave (PAW) method was used to replace the pseudopotential of inner core electrons and nucleus with the valence electrons[58]. The Kohn-Sham electron wave functions were expanded using the plane-wave functions with an energy cutoff of 400 eV. The optimization was considered convergence and each atom would be fully relaxed until the spring force between adjacent images was less than 0.05 eV Å$^{-1}$. The total energy change upon two steps for the electronic self-consistent field iteration was less than 1E−4 eV.

Single-layer HsGDY and GDY was simulated by a repeated slab model with a 1 × 1 supercell. For the HsGDY (Supplementary Fig. 2a), the calculated lattice constants are a = b = 16.38 Å and c = 20.00 Å with α = β = 90 and γ = 120. For the GDY (Supplementary Fig. 2b), the calculated lattice constants are a = b = 9.46 Å and c = 20.00 Å with α = β = 90 and γ = 120. For the graphene (Supplementary Fig. 2c), the calculated lattice constants are a = b = 9.87 Å and c = 20.00 Å with α = β = 90 and γ = 120. For SWCNT (Supplementary Fig. 2d), the calculated lattice constants are a = b = 32.00 Å and c = 4.92 Å with α = β = 90 and γ = 120. The diameter of the SWCNT is 1.49 nm (consistent with experiments) and the spiral direction is along the c-axis. The replicas of layers were separated by a vacuum layer of 15 Å along the z-direction, which led to negligible interactions between the research system and their mirror images. Meanwhile, the 3 × 3 × 1 k-point meshes in Brillouin zone was sampled for structure optimization and electronic property according to the Monk Horst–Pack scheme[59]. As for the adsorption case of atoms and bulk, the binding energy ($E_f$) was defined as

$$E_f = E_{Hg-HsGDY} - \left( E_{HsGDY} + nE_{Hg} \right) \qquad (1)$$

where $E_{HsGDY}$, $E_{Hg}$, and $E_{HsGDY–Hg}$ are the total energies of the bulk HsGDY substrates before formation and isolated free Hg atoms in its bulk form, and the total energy of Hg atoms inserted in HsGDY monolayer, respectively. All three types of energies were derived from the self-consistent field calculations using the same calculated setting parameters. With this definition, a negative value indicates an exothermic adsorption. The more negative this value is, the more stable configuration has been proved.

Charge transfers were calculated using the Bader charge analysis method. In addition, the climbing images nudged elastic band (CL-NEB)[59] and dimer method[60] were used to carry out the transition state (TS), which is further verified by means of frequency calculations. The number of inserting image was chosen by the formula "dist/0.8" derived from the difference-comparing scripts called dist.pl embedded in the transition state tools (VTST) software package compiled in VASP.

The Ab initio molecular dynamics (AIMD) simulations were performed under NVT canonical ensemble with a target temperature by the Nose-Hooverthermostat. The Verlet algorithm were used for integration and the time-step was 2 fs. The simulation time for each MD trajectories was 7 ps and the thermodynamic quantities of the system were statistically averaged in the last 2 ps.

The relative Gibbs energy of Hg as well as HsGDY with regard to the reaction of $H^+ + e^- \rightarrow 1/2\, H_2$ was calculated to compare their abilities to lose electrons (Supplementary Fig. 22). The Gibbs free energy ($\Delta G_{H^*}$) is defined as follows: $\Delta G_{H^*} = \Delta E_{H^*} + \Delta E_{ZPE} - T\Delta S$[61], where $\Delta E_{H^*}$, $\Delta E_{ZPE}$, $\Delta S$ are the adsorption energy of hydrogen atom on surface, the zero point energy and the entropy change after the adsorption of hydrogen atom, respectively. $\Delta E_{H^*}$ is calculated by $\Delta E_{H^*} = E_{H/slab} - E_{slab} - 1/2E_{H2}$, where $E_{H/slab}$, $E_{slab}$ are the energies of hydrogen atom adsorbed model, bare model, and $E_{H2}$ is the energy of hydrogen molecule isolated in vacuum. The zero-point energy change $\Delta E_{ZPE}$ is obtained from vibrational

frequency calculation. The entropy ($S_O$) of the molecular hydrogen is used in the gas phase at standard conditions (1 bar of $H_2$, pH = 0, and temperature $T = 300$ K). The entropy of the adsorbed hydrogen atom is negligible because the hydrogen atom is bound to the surface. Hence, the $\Delta S$ can be estimated by $-1/2 \times S_O$, and $T\Delta S$ is about $-0.2$ eV at $T = 300$ K according to the reported literature[62].

**Comparing the charge non-uniformity of GDY and HsGDY.** According to the electron localization function (ELF) results as shown in Fig. 1b and Supplementary Fig. 1b, we quantitively compared the charge non-uniformity of GDY and HsGDY by borrowing the concept of dispersion degree and also considering the mercury binding sites (the center of triangular hole for GDY and the side sites of acetylenic bond in plane for HsGDY). A high ELF value (0–1) implies a high probability of electron localization. The sampling points are uniformly located at the circle with radius of $r$ and centered in the mercury binding site as shown in Supplementary Fig. 3. The calculation method is as follows:

$$\sigma_r = \sqrt{\frac{\sum_{i=1 \to n}(X_{i,r} - X_r)^2}{n-1}} \qquad (2)$$

where $\sigma_r$ indicates the charge non-uniformity; $X_{i,r}$ represents the ELF value of sampling point; $X_r$ represents the average ELF value of sampling points; $r$ denotes the distance from the mercury binding site to the sampling point; n denotes the number of sampling points.

**Adsorption performance evaluation.** The adsorption performances of gas-phase $Hg^0$ over different carbon materials were examined by a laboratory-scale fixed-bed reaction system as shown in Supplementary Fig. 5. A constant $Hg^0$ vapor (340 μg m$^{-3}$) was produced from a mercury permeation device. The generated $Hg^0$ vapor was mixed with other gas components and then introduced into a quartz tube reactor where 100 mg sample was placed. The tube reactor was wrapped in a tubular furnace to maintain a desired reaction temperature. With $N_2$ carrying, $O_2$ (5% vol.), $SO_2$ (600/1200/1800/2400 ppm), NO (600/1000 ppm), and $H_2O$ (1/3% vol.) were added into the gas stream when needed. All gas flows were controlled by flowmeters, with a total flow rate of flue gas maintained at 400 ml min$^{-1}$. The gas hourly space velocity (GHSV) of the experiment was ~53000 h$^{-1}$. An online mercury analyzer (QM201H, Suzhou Qing'an Instrument Co., Ltd) was used to detect the inlet and outlet $Hg^0$ concentrations. Gases containing $SO_2$/NO and water vapor were purified by $Na_2O_2$ and silica gel before entering the mercury analyzer. For each test, the gas stream was first switched to bypass and the inlet gas was sampled to acquire stable $Hg^0$ concentrations. Thereafter, the $Hg^0$ containing gas flow was passed through the reactor for $Hg^0$ adsorption tests. At the end of test, the gas stream was switched to bypass again to verify the stability of feed $Hg^0$ concentration. $Hg^0$ and acid gases in tail gas were removed by solutions of acidic potassium permanganate and sodium hydroxide. The following equations were employed to evaluate the $Hg^0$ removal efficiency $\left( E_{Hg} \right)$ and the $Hg^0$ adsorption capacity ($q_t$), respectively.

$$E_{Hg} = \left( 1 - \frac{Hg^0_{out}}{Hg^0_{in}} \right) \times 100\% \qquad (3)$$

$$q_t = \frac{1}{m} \int_0^t (Hg^0_{in} - Hg^0_{out}) \times F \times dt \times 10^{-3} \qquad (4)$$

where $E_{Hg}$ indicates the $Hg^0$ removal efficiency; $Hg^0_{in}$ and $Hg^0_{out}$ represent the instantaneous concentrations of $Hg^0$ at the inlet and outlet of the reactor, respectively, μg m$^{-3}$; $m$ denotes the weight of sorbent, g; $F$ represents the gas flow rate, m$^3$ min$^{-1}$; $t$ denotes the adsorption time, min; $q_t$ is the $Hg^0$ adsorption capacity, mg g$^{-1}$.

**Hg-TPD analysis.** To identify the formed mercury species on HsGDY, the temperature programmed desorption of Hg (Hg-TPD)

from the spent HsGDY was carried out. The Hg-laden HsGDY in the quartz tube reactor was firstly purged by $N_2$ (400 ml min$^{-1}$) at room temperature for 20 min. Then the sample was heated from 20 °C to 400 °C at a ramping rate of 2 °C min$^{-1}$ in $N_2$ atmosphere. During this process, the desorbed mercury concentration was measured by a mercury analysis device (QM201H, Suzhou Qing'an Instrument Co., Ltd).

Identification of the desorbed mercury species by OHM. The OHM was employed to examine the mercury speciation distribution during thermal regeneration. KCl solution (1 M) and 3% (w/v) KMnO$_4$ in 10% (v/v) H$_2$SO$_4$ solution were used to capture the oxidized mercury and elemental mercury in gas stream, respectively. The resulting Hg concentration was measured by using Cold-Atomic Fluorescence Spectroscopy (AFS-930, Beijing Titan Instruments Co., Ltd).

Mercury desorption by acid washing. Before the acid-washing experiment, the mercury-laden HsGDY was purged by $N_2$ for 30 min. Then the mercury-laden HsGDY was desorbed using 15 ml HNO$_3$ solutions under continuously stirring for 24 h. Thereafter, the HsGDY was separated by filtration, and the mercury concentration in the filtrate was measured by ICP-OES (Prodigy 7, USA). Additionally, the pH of the obtained solution is adjusted to 7–8 by NaOH. Afterwards, Na$_2$S dilute solution is dropwisely added into the aforementioned solution under continuously stirring at room temperature.

Kinetic simulation. The kinetic simulation can help to understand the Hg$^0$ adsorption mechanism. The kinetic equations of the pseudo-first-order model, pseudo-second-order model, and Weber–Morris model are illustrated as follows:

$$q_t = q_e \times \left(1 - e^{-k_1 t}\right) \tag{5}$$

$$q_t = \frac{t}{\frac{1}{k_2 q_e^2} + \frac{t}{q_e}} \tag{6}$$

$$q_t = k_{id} t^{1/2} + C \tag{7}$$

where $q_t$ and $q_e$ are defined as the Hg$^0$ adsorption capacities at time t and at equilibrium, respectively, mg g$^{-1}$; $k_1$ is the adsorption rate constant for the pseudo-first-order model, min$^{-1}$; $k_2$ represents the rate constant for the pseudo-second-order model, g mg$^{-1}$ min$^{-1}$; $k_{id}$ represents the intra-particle diffusion coefficient or the adsorption rate constant for the model, mg g$^{-1}$ min$^{-1/2}$); constant $C$ is associated with the thickness of the boundary layer, mg g$^{-1}$; $t$ represents the adsorption time, min. By nonlinear fitting the adsorption break-through curve data to the models, the values of $k_1$, $k_2$, $k_{id}$ and $q_e$ can be obtained.

Demonstrating the oxidation of Hg over HsGDY by electrochemical experiments. A standard three-electrode system was employed to conduct the electrochemical experiments at a CHI660E electrochemical workstation. The clean glassy carbon electrode as cathode (Pt as anode) was immersed in $1.5 \times 10^{-3}$ M Hg(NO$_3$)$_2$ and 0.1 M KCl solutions. Then Hg will be deposited on the glassy carbon electrode to form an as-prepared Hg electrode by electrolysis. The HsGDY/Hg electrode (Hg deposited on HsGDY) was prepared from HsGDY electrode as a cathode by the same electrolysis method. The HsGDY electrode, Hg electrode, and HsGDY/Hg electrode were employed as the working electrodes, respectively. The Pt and saturated calomel electrodes were used as auxiliary electrodes and reference electrodes, respectively. The electrolyte was 0.5 M Na$_2$SO$_4$ solution. The electrochemical cyclic voltammetry (CV) testing was performed within the range of −1.5 to 1.5 V. Meanwhile, the CV characteristic curve was obtained and the results are displayed in Supplementary Fig. 23.

## Data availability
The data supporting the findings of the study are included in the main text and supplementary information files. Raw data are available from the corresponding author upon request.

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

## Acknowledgements

This work was supported by the National Natural Science Foundation of China (22076060—Y.G., 51702112—Y.G., 52100134—H.L.), the Open Foundation of the Project of the State Key Laboratory of Advanced Technology for Materials Synthesis and Processing, Wuhan University of Technology, China (2020-KF-17—Y.G.), the Recruitment Program of Global Young Experts Start-up Funds of China, the Program of Introducing Talents of Discipline to Universities of China (B17019—Y.G.), Knowledge Innovation Program of Wuhan Shuguang Project (2022010801020289—Y.G.), the China Postdoctoral Science Foundation (2022M721284—H.L.) and the Project of the Hubei International Scientific and Technological Cooperation Base of Pesticide and Green Synthesis (Y.G.).

## Author contributions

H.L., C.P., and Y.G. designed the experiments, analyzed the data, and wrote the manuscript. X.P. evaluated the adsorption performance and analyzed the data. B.Z. carried out the DFT calculations and AIMD simulations. S.S., Z.X., and X.Q. synthesized the sorbent. X.Q. and C.P. carried out the material characterization experiments. Y.L. and J.W. conducted the electrochemical experiments and analyzed the results. Y.G. supervised the work. All the authors contributed to the overall scientific interpretation and edited the manuscript. H.L. and C.P. contributed equally.

## Competing interests

The authors declare no competing interests.
