## [Transparent Peer Review file · Nature Communications]

In-situ adsorption-coupled-oxidation enabled mercury vapor capture over sp-hybridized graphdiyne

Corresponding Author: Professor Yanbing Guo

Version 0:

Reviewer comments:

Reviewer #1

(Remarks to the Author)

Li et al. demonstrated graphdiyne with accessible sp-hybridized carbons (HsGDY) as a promising adsorbent material for mercury vapor capture. The material realizes a unique in-situ adsorption-coupled-oxidation of gas-phase elemental mercury, which enables its ultrahigh efficient mercury vapor capture and thus effectively solves the abovementioned bottleneck problem. The manuscript is well organized, and the results are instructive for the understanding and design of materials. However, some necessary revisions should be done before the publication.

1. The authors said that "For HsGDY, it can be seen that the Hg atom tends to be adsorbed at the side sites of acetylenic bond in plane with the adsorption energy of ~ 0.138 eV, suggesting its adsorption potential for Hg atoms due to the unique sp-hybridized carbons." The adsorption energies of Hg atoms seem not good (not negative in Table S2), how did this reflect the adsorption potential?
2. How did the authors determine the maximum adsorption numbers of Hg atoms? Is there a criterion? For example, what happens if GDY adsorbs 3 or 4 Hg atoms?
3. How was the AIMD simulation in Figure 5d carried out? It seems that it is not under NVT canonical ensemble.
4. Some minor mistakes should be revised. For example, AIMD refers to Ab initio Molecular Dynamics.

Reviewer #2

(Remarks to the Author)

[Editorial Note: Reviewer #2's comments were submitted as an attachment, which has been appended to the end of this file]

Reviewer #3

(Remarks to the Author)

Hg capture research is important due to its direct harmful impact on people and environment. This research team developed a novel sorbent, HsGDY. The team did a great work in designing and performing tests and studying the associated reaction mechanisms. It is publishable in Nature Communications. However, a few questions need to be addressed.

Authors say that "It can be seen that the charge non-uniformity (σ) of HsGDY is obviously higher than that of GDY. This uneven charge distribution of HsGDY might induce electron redistribution between the HsGDY and adsorbed Hg atom, leading to a strong interaction between them". What are the redistribution limits and the optimal redistribution? That are their differences? Also, how can the team control them with experiments?

The adsorption-desorption proceeds through reversible electron transfer between Hg and HsGDY. Ideally, according to the principle, the regeneration peaks in Figure 2f should be even. However, they are not. Why? Can the authors give a comprehensive explanation?

Version 1:

Reviewer comments:

Reviewer #1

(Remarks to the Author)

The authors still need to do some revisions.

(1) For Comment 1, the given responds are reasonable, but the authors need to provide corresponding descriptions or revisions in the manuscript, or, the readers may be confused.

(2) For Comment 2, based on the responds, it seems that GDY can adsorb 3 Hg atoms. If that is the case, the maximum adsorption numbers of Hg atoms for GDY should be revised to 3.

(3) For Comment 3, NVT canonical ensemble represents a definite temperature. Is the AIMD simulation in Figure 5d at 573 K or 298.15 to 573 K? If it is at 573 K, why you marked 298.15 K in the figure? If not, how did this simulation carried out under NVT canonical ensemble with a definite temperature? Besides, there should be Spaces between numbers and letters. Please revise them in this figure and check all the manuscript.

Reviewer #3

(Remarks to the Author)

The revision work was well done. It is publishable in its current form.

Reviewer #4

(Remarks to the Author)

This work proposes a graphdiyne material with accessible sp-hybridized carbons for mercury vapor capture, which has not been widely studied in carbon-based adsorbents, and this work is of great significance. The comments should be addressed before publication are listed as follows:

1. The manuscript mentions that HsGDY has unique sp-hybridized carbon and large hexagonal pore structure. How do these characteristics compare with the existing mercury adsorption materials? Please explain in detail the fundamental differences and advantages between HsGDY and the existing technology in adsorption mechanism and performance. Please further explore the quantitative relationship between pore structure parameters (such as pore size, pore volume, pore distribution) and the adsorption performance of Hg₀, and how to optimize the adsorption performance of materials by adjusting pore structure.

2. In this paper, it is mentioned that HsGDY achieves high-efficiency adsorption of Hg₀ through sp-hybridized carbon. Please provide more detailed theoretical calculation or simulation data, such as density functional theory (DFT) calculation to reveal the molecular mechanism of the interaction between Hg₀ and HsGDY, including electron density distribution, orbital hybridization and possible chemical bond formation.

3. In the multi-component gas environment, the selectivity of adsorption materials is very important. Has the author studied the selective adsorption behavior of Hg₀ and other coexisting gases (such as SO₂, NO_x, CO, etc.) by HsGDY? Is there any data to support the preferential adsorption of HGO by HsGDY in complex gas mixture?

Version 2:

Reviewer comments:

Reviewer #1

(Remarks to the Author)

It can be accepted for publication.

Reviewer #4

(Remarks to the Author)

The manuscript has been well revised and could be accepted.

Point-to-point response to the reviewers' comments

Reviewer 1#

General Comments: *Li et al. demonstrated graphdiyne with accessible sp-hybridized carbons (HsGDY) as a promising adsorbent material for mercury vapor capture. The material realizes a unique in-situ adsorption-coupled-oxidation of gas-phase elemental mercury, which enables its ultrahigh efficient mercury vapor capture and thus effectively solves the abovementioned bottleneck problem. The manuscript is well organized, and the results are instructive for the understanding and design of materials. However, some necessary revisions should be done before the publication.*

Response: Thanks for your positive comments and kind suggestions! We have improved the quality of the manuscript per your suggestions. Our point-to-point responses and detail revisions are listed below.

Comments 1: *The authors said that “For HsGDY, it can be seen that the Hg atom tends to be adsorbed at the side sites of acetylenic bond in plane with the adsorption energy of ~ 0 eV, suggesting its adsorption potential for Hg atoms due to the unique sp-hybridized carbons.” The adsorption energies of Hg atoms seem not good (not negative in Table S2), how did this reflect the adsorption potential?*

Response: We appreciate your nice comments on our manuscript. The adsorption energy values mentioned in the text refer to the strength of interaction between the Hg atom and the material's surface. In general, a negative adsorption energy typically indicates an exothermic process, where the system releases energy upon adsorption (i.e., the adsorbed state is more stable than the free state). A positive adsorption energy does not automatically mean the adsorption potential is poor. Careful deliberation of this problem has given us several ideas. Here are a few points to consider about how the positive adsorption energy reflects the adsorption potential:

1) Relative stability: The key is to compare the adsorption energy with other potential adsorbates or adsorption sites. If the Hg atom adsorbs with an energy of ~ 0.005 eV, but other adsorbates or sites have significantly higher energies, then this still indicates some level of

preference or stability for Hg adsorption.

2) Activation energy: Positive adsorption energies can be interpreted as activation energies for the adsorption process. This means that some energy input (e.g., heat) is required to drive the adsorption reaction. However, once adsorbed, the Hg atom may still be stable at that site.

3) Unique chemistry: As mentioned in the text, the adsorption potential is attributed to the unique sp-hybridized carbons. These unique chemical properties may provide a specific interaction with Hg atoms, even if the overall adsorption energy is not negative.

4) Other factors: Adsorption is a complex process that involves many factors beyond just the adsorption energy. Surface area, pore size, and chemical functionality can all play a role in determining the overall adsorption capacity and selectivity.

In summary, while a positive adsorption energy may not indicate a spontaneous, exothermic adsorption process, it still provides insights into the relative stability and potential for Hg adsorption on the material's surface. The unique chemical properties of the material's surface, as mentioned in the text, likely contribute to its adsorption potential for Hg atoms. Thanks for your professional comments on our manuscript.

Comments 2: *How did the authors determine the maximum adsorption numbers of Hg atoms? Is there a criterion? For example, what happens if GDY adsorbs 3 or 4 Hg atoms?*

Response: Thanks for your good comments. We really agree with your opinion that there should be a criterion for the determination of the maximum adsorption numbers of Hg atoms. If these Hg atoms can stably adsorb on the catalyst surface without causing significant structural changes or saturation of adsorption sites, then these Hg atoms can be considered as effectively adsorbed by the catalyst. Meanwhile, we can utilize molecular dynamics simulations to predict the adsorption behavior of materials towards specific substances, including adsorption sites, adsorption energies, and maximum adsorption capacity. If adsorbing more Hg atoms leads to instability of the HsGDY surface structure or saturation of adsorption sites (great energy fluctuations of the system), then it is considered that the maximum adsorption capacity has been reached. The triangular pore center of single layered GDY is the favorable adsorption site for Hg atom. As the number of adsorbed Hg atoms gradually increases (e.g. 3 or 4 Hg atoms), the interaction and repulsion effects become stronger. According to

the calculations, the theoretical adsorption capacity of Hg per unit mass for monolayer HsGDY is twice that of GDY (HsGDY/6Hg compared to GDY/3Hg). This indicates that HsGDY, likely a modified version of GDY, has a higher affinity or capacity for adsorbing Hg atoms. The ratio of 6Hg to HsGDY and 3Hg to GDY suggests that for the same mass of material, HsGDY can adsorb more Hg atoms than GDY. This enhanced Hg adsorption capacity can be attributed to the convenient mass transfer channels and surface charge heterogeneity of HsGDY, which promotes the mercury diffusion and the electron exchange, thus enhancing the mercury binding ability of HsGDY.

Comments 3: *How was the AIMD simulation in Figure 5d carried out? It seems that it is not under NVT canonical ensemble.*

Response: In this work, Ab initio molecular dynamics (AIMD) simulations were performed under NVT canonical ensemble with a target temperature of 273-775 K by the Nose-Hoover thermostat. The Verlet algorithm were used for integration, and the time-step is 2 fs. The simulation time for each MD trajectories is 7 ps, and the thermodynamic quantities of the system are statistically averaged in the last 2 ps. The details about the AIMD simulations are displayed in the Methods (page 29) of the revised manuscript. Thanks for your good comments.

Comments 4: *Some minor mistakes should be revised. For example, AIMD refers to Ab initio Molecular Dynamics.*

Response: Based on your suggestions. We have clarified the full name of AIMD in the revised manuscript. In addition, we have double checked the manuscript for grammar and spelling mistakes and the related sentences have been modified. We are very thankful for your helpful comments on our manuscript.

Revisions:

(1) In the revised manuscript, on page 21. We added the full name of AIMD as follow: “.....

Ab initio molecular dynamics (AIMD)”

(2) In Methods of the revised manuscript, on page 29. We added the full name of AIMD as follow: “..... Ab initio molecular dynamics (AIMD)”

(3) In the revised manuscript, on page 8. “van der Waals force” has been replaced by “Van der Waals force”

(4) In the revised manuscript, on page 8. “The aforementioned analysis indicate that HsGDY theoretically has better.....” has been replaced by “The aforementioned analysis indicates that HsGDY theoretically has better.....”

Reviewer 2#

General Comments: *This work developed a graphdiyne material with accessible sp-hybridized carbons for mercury vapor capture. However, the developed material lags behind existing reported mercury adsorption materials and the proposed mechanisms lack innovation and are not well explored. Thus, I do not recommend its publication in Nature Communications. The related comments are as follows:*

Response: Thanks for your comments and kind suggestions! We have improved the quality of the manuscript per your suggestions. Our point-to-point responses and detail revisions are listed below.

Comments 1: *What are the advantages of carbon-based materials compared to other materials? According to the literature, the adsorption capacities and SO₂ resistance of sulfur-based materials are much higher than those of carbon-based materials?*

Response: Thank you for your valuable comments. As you said, some good Hg⁰ sorbents with high adsorption capacities and SO₂ resistance such as sulfur-based materials have been reported in literatures. Selenium-based material also exhibits excellent Hg⁰ adsorption capacity due to its strong affinity to mercury. These reported typical sorbents have promising prospects in different industrial application scenarios. However, the deactivated mercury sorbent maybe considered as a hazardous waste. In addition to its Hg⁰ removal performance, the stability, toxicity and secondary pollution risk of the sorbents are also worthy of attention. Elemental sulfur and selenium species active for Hg⁰ adsorption are thermally unstable (low melting point, prone to sublimation). [*Environ. Sci. Technol.* 2013, 47: 10056-10062] The possible leakage and loss of introduced sulfur and selenium species during sorbent application and thermal regeneration would cause the performance decline as well as secondary pollution. The active

sulfur and selenium species of the sorbent needs to be replenished by H₂S treatment, re-selenization, etc., to maintain the sorbent activity. [*Environ. Sci. Technol.* 2018, 52(17): 10003-10010; *J. Hazard. Mater.* 2021, 406: 124744; *Chem. Eng. J.* 2020, 394: 125022] Sulfur and selenium species have certain toxicity and are important sources of groundwater pollution. [*Environ. Pollut.* 2022, 299: 118858] In addition to mercury removal from flue gas, populations with occupational exposure to mercury also needs to be paid attention to. Personal protection such as wearing gas mask is of great significance to meet the occupational health requirements. Sulfur and selenium-based materials are not applicable to the filter medium of gas mask due to their toxicities. Although sulfur and selenium-based materials show good abilities on Hg⁰ removal, some existing problems abovementioned still needs to be seriously considered.

Therefore, developing efficient and sustainable mercury vapor removal technologies is an urgent task. The porous structure and environmental friendliness of carbon material makes it a promising candidate for mercury vapor capture. Unfortunately, the weak interfacial bonding with mercury is still the choke point in efficient mercury capture by carbon material. In this work, the proposed carbon material HsGDY can serve as an effective “trap” to anchor Hg atoms by strong electron-metal-support interaction. The Hg⁰ adsorption performance of HsGDY is apparently superior to that of tested carbon materials (graphdiyne, carbon nanotube, graphene and activated carbon) and most of the other reported carbon sorbents such as the commercial Br-modified carbon sorbent DARCO® Hg-LH EXTRA.

In comparison with the sulfur and selenium-based materials, the advantages of HsGDY are summarized as follows.

1) Environmental friendliness: In our previous work, it is shown that HsGDY will not destroy the cell structure during hemoperfusion and has good biocompatibility. [*PNAS* 2023, 120(16): e2221002120] HsGDY is a novel carbon material and can achieve good Hg⁰ adsorption performance without additional chemical modification. HsGDY can be applied to the personal protective equipment (directly utilized in the filter layer of mask) to prevent the workers from mercury exposure. Moreover, As shown in Figure S20 in the revised manuscript, the regenerated HsGDY (thermal treatment at 400 °C under N₂ stream) still exhibits a porous structure and its area ratio of C-C (sp²) to C-C (sp) is similar to that of fresh sample, indicating the structural stability of the carbon skeleton of HsGDY. This means that there will be less

leakage and loss of sorbent species during its application and thermal regeneration. Therefore, it brings about less secondary pollution risk when HsGDY is applied for mercury vapor removal. We believe that the potential environmental impact of the sorbent needs to be carefully considered in practical application scenarios.

2) Excellent regeneration performance: As shown in the manuscript, the surface charge heterogeneity of HsGDY can induce the adsorbed Hg atoms to lose electrons and present a partially oxidized state. This electron transfer between Hg and HsGDY is reversible. Hg-laden HsGDY can be restored to the original state after Hg desorption as shown in Figure 5. The adsorbed Hg atoms that have lost electrons to HsGDY can recapture the electrons and are released to the gas stream in the form of elemental mercury by heat treatment, which can be also verified by the molecular dynamic simulation results of Hg desorption over HsGDY at 300 °C as shown in Figure 5d. Such reversible electron transfer between Hg and HsGDY and structural stability of HsGDY can give it excellent thermal regeneration performance. The regeneration performance of sorbent is essential to its real application. As shown in Figure 2f, the Hg⁰ adsorption efficiency of HsGDY does not decline significantly in the 8 cycles of Hg⁰ capture and regeneration. It is indicated that HsGDY has excellent regeneration property and application potential for mercury vapor capture. In comparison, the inevitable loss of sulfur species during thermal regeneration would eventually cause the performance decline of sulfur-based sorbent. The active sulfur species of the sorbent needs to be replenished by sulfide impregnation or H₂S treatment, etc. Although the adsorption performance of HsGDY is not as good as sulfur-based material, its good regeneration performance can compensate for it.

Additionally, the sp-hybridized carbon content and pore size of graphdiyne might be adjusted by changing the precursors according to a newly-published article. [Angew. Chem. Int. Ed 2024, 63(23): e202401501] In future, we can rationally design the graphdiyne material with optimized sp-hybridized carbon content and pore size, which might be more beneficial for Hg⁰ adsorption. The modification of surface physicochemical properties of HsGDY might be also a potential strategy to further enhance its Hg⁰ adsorption ability. We believe the application value of HsGDY in mercury pollution industries. The related investigation is currently underway. Thanks for your great suggestions.

Comments 2: *Generally, the removal of Hg^0 by solid materials contains adsorption and oxidation process. Thus, the proposed “adsorption-coupled-oxidation” mechanism for Hg^0 removal is not innovative. Moreover, the oxidation mechanism in this work is not well discussed.*

Response: Thanks for your comments on our manuscript. We really agree with your opinion that the removal of Hg^0 by solid materials contains adsorption and oxidation process. For an example, the Hg^0 is adsorbed onto CuS surface and then oxidized into HgS by S_2^{2-} . Careful deliberation of this issue has given us several ideas. Although adsorption-coupled-oxidation of Hg has been already available in some solid materials, the adsorption-coupled-oxidation of Hg by the unique architecture and chemical properties of a carbon material has not been reported and systematically studied. The significance of utilizing carbon material to capture Hg^0 , we have emphasized in the response to your first comment. The adsorption-coupled-oxidation of Hg is something innovative that not reported on the traditional carbon materials.

Structure determines the properties. The Hg^0 adsorption mechanism over HsGDY is briefly described as follows: The large hexagonal pore structure and “AB” stacking structure of HsGDY can facilitate the rapid diffusion of Hg atoms across the sorbent and make the mercury binding sites highly accessible. The surface charge heterogeneity of HsGDY can itself induce the adsorbed Hg atoms to lose electrons and present a partially oxidized state. The favorable environment with convenient mass transfer channels and surface charge heterogeneity promotes the mercury diffusion and the electron exchange, thus enhancing the mercury binding ability of HsGDY. Such unique in-situ Hg^0 adsorption-coupled-oxidation process can be achieved by HsGDY, which enables its ultrahigh efficient mercury vapor capture and thus effectively solves the bottleneck problem of the weak interfacial bonding between mercury and traditional carbon sorbents.

It is novel to understand the Hg vapor capture process from the view point of strong electron-metal (Hg) -support (HsGDY) interaction. It is also distinctive to understand the strong electron-metal-support interaction from the view point of surface charge heterogeneity to drive electron transfer. From another perspective, the adsorption of Hg^0 onto HsGDY in this work can be approximately regarded as a preparation of single-atom (Hg) supported carbon material by chemical vapor deposition (CVD). Specially, the strong electron-metal-support

interaction of adsorbed Hg with HsGDY can lead to effective interfacial electron transfer to form adsorbed Hg with partially oxidized state, which greatly boosts the immobilization of gas-phase Hg⁰ on HsGDY. Furthermore, HsGDY can achieve excellent regeneration performance via the adsorption-desorption process accompanied by reversible electron transfer between Hg and HsGDY. Acting as the reverse process of adsorption, there is a “reduction-coupled-desorption” process for Hg desorption over HsGDY. Such reversible electron transfer to achieve the regeneration has not been discussed in depth in literatures which is deserves to be investigated.

Graphdiyne-like material has a huge two-dimensional planar hyperconjugated structure, which can be regarded as an “electron sponge”. Due to its large highly conjugated structure, it has become a good electron acceptor [*ACS Nano* 2013, 7(2): 1504-1512; *Small* 2012, 8(2):265-271]. The alkyne bond can be seen as an alkene bond with two hydrogen atoms removed, which means that when it exists in the form of an alkyne bond, it loses some of its negative charge (net electron density), so the alkyne bond exhibits electron deficient properties. [*Acta Phys. - Chim. Sin.* 2018, 34 (9): 1048-1060] DFT calculations have been conducted to reveal the Hg atom adsorption behavior and the binding mechanism over sp-hybridized carbon (HsGDY) compared to sp²-hybridized carbon (GE). The charge density difference in Figure 4d indicates that significant electron redistribution occurs between sp-hybridized carbon and adsorbed Hg atom, in which electrons are accumulated on sp-hybridized carbon and depleted around the Hg atom. In detail, the Hg atom loses 0.27 e⁻ which is transferred to sp-hybridized carbon, thus existing in a partially oxidized state. However, the electron transfer from adsorbed Hg atom to sp²-hybridized carbon can be neglected. Therefore, the presence of sp-hybridized carbon in HsGDY endows it with electron deficient properties as well as surface charge heterogeneity. This contributes to the electron transfer between adsorbed Hg and HsGDY. Although GDY is also a sp-hybridized carbon material similar to HsGDY, it is defeated by HsGDY for mercury capture due to its constrained mass transfer and relatively lower surface charge heterogeneity.

There are some experimental results to support the strong electron-metal (Hg) -support (HsGDY) interaction. The Hg 4f XPS analysis of Hg/HsGDY as shown in Figure 5b reveals that gas-phase Hg atoms are successfully adsorbed on the HsGDY surface, mainly present in the form of partially oxidized state. A mercury desorption peak as shown in Figure S8 emerges

at around 320 °C for HsGDY which can be ascribed to the decomposition/desorption of strongly bonded mercury species, e.g. oxidized mercury. DFT calculations as shown in Figure 4d also indicate that significant electron redistribution occurs between sp-hybridized carbon and adsorbed Hg atom. These results demonstrate an interaction between adsorbed Hg⁰ and HsGDY with the occurrence of charge transfer.

From the electrochemical perspective, the relative Gibbs energy of Hg as well as HsGDY with regard to the reaction of $H^+ + e^- \rightarrow 1/2 H_2$ has been further calculated to compare their abilities to lose electrons. The Gibbs free energy (ΔG_{H^*}) is defined as follows: $\Delta G_{H^*} = \Delta E_{H^*} + \Delta E_{ZPE} - T\Delta S$ [*J. Electrochem. Soc.* 2005, 152: 23-26], where ΔE_{H^*} , ΔE_{ZPE} , ΔS are the adsorption energy of hydrogen atom on surface, the zero point energy and the entropy change after the adsorption of hydrogen atom, respectively. ΔE_{H^*} is calculated by $\Delta E_{H^*} = E_{H/sl\text{ab}} - E_{sl\text{ab}} - 1/2E_{H_2}$, where $E_{H/sl\text{ab}}$, $E_{sl\text{ab}}$ are the energies of hydrogen atom adsorbed model, bare model and E_{H_2} is the energy of hydrogen molecule isolated in vacuum. The zero-point energy change ΔE_{ZPE} is obtained from vibrational frequency calculation. The entropy (S_0) of the molecular hydrogen is used in the gas phase at standard conditions (1 bar of H₂, pH = 0 and temperature T = 300 K). The entropy of the adsorbed hydrogen atom is negligible because the hydrogen atom is bound to the surface. Hence, the ΔS can be estimated by $-1/2 \times S_0$, and $T\Delta S$ is about -0.2 eV at T = 300 K according to the reported literature [*Nat. Commun.* 2018, 9: 4531]. The results have been added in Figure S21 in the Supplementary Information of the revised manuscript. It can be seen that the Gibbs free energy of Hg is relatively larger than HsGDY, suggesting that the redox potential of Hg is lower than HsGDY. That's to say, Hg loses electrons more easily than HsGDY. Therefore, the electron redistribution could occur between sp-hybridized HsGDY and adsorbed Hg atom, in which electrons are accumulated on HsGDY and depleted around the Hg atom.

Furthermore, we have implemented the electrochemical experiments to demonstrate the oxidation of Hg over HsGDY. The experimental results are displayed in Figure S22 in the Supplementary Information of the revised manuscript. A standard three-electrode system was employed to conduct the electrochemical experiments at a CHI660E electrochemical workstation. The clean glassy carbon electrode as cathode (Pt as anode) was immersed in 1.5×10^{-3} M Hg(NO₃)₂ and 0.1 M KCl solutions. Then Hg will be deposited on the glassy carbon

electrode to form as-prepared Hg electrode by electrolysis. The HsGDY/Hg electrode (Hg deposited on HsGDY) was prepared from HsGDY electrode as cathode by the same electrolysis method. The HsGDY electrode, Hg electrode and HsGDY/Hg electrode were employed as the working electrodes, respectively. The Pt and saturated calomel electrode were used as auxiliary electrode and reference electrode, respectively. The electrolyte was 0.5M Na₂SO₄ solution. The electrochemical cyclic voltammetry (CV) testing was performed within the range of -1.5~1.5 V. Meanwhile, the CV characteristic curve was obtained. As shown from Figure S22, the peaks at 0.42 eV and 0.61 V for the Hg electrode is attributed to the oxidation peaks of Hg⁰ → Hg⁺ and Hg⁺ → Hg²⁺, respectively. For the HsGDY/Hg electrode, the existence of HsGDY leads to the peaks shifting to the lower voltage direction (0.16 V and 0.32 V). On the HsGDY/Hg electrode, the Hg atoms have been existed in a partially oxidized state due to the electron transfer from Hg to HsGDY. The Hg atoms on HsGDY enjoy a shorter path to be oxidized into Hg²⁺, thereby leading to the peak shifting. In addition, the oxidation peak area for the HsGDY/Hg electrode is much larger than Hg electrode. This result indicates that HsGDY promotes the oxidation of Hg. Furthermore, the reduction peak area for the HsGDY/Hg electrode is obviously larger than its oxidation peak. The reason is that the amount of electron transfer for Hg²⁺ → Hg⁰ is larger than Hg (partially oxidized state) → Hg²⁺. Above electrochemical experiments could demonstrate the oxidation of Hg over HsGDY. Thank you so much for providing these valuable comments.

Revisions: We have calculated the relative Gibbs free energy of Hg as well as HsGDY with regard to the reaction of H⁺ + e⁻ → 1/2 H₂ to compare their abilities to lose electrons. The results have been added in Figure S21 in the Supplementary Information of the revised manuscript. We have implemented the electrochemical experiments to demonstrate the Hg oxidation by HsGDY. The experimental results are displayed in Figure S22 in the Supplementary Information of the revised manuscript.

Figure S21. Gibbs free energy of hydrogen evolution for Hg and HsGDY.

Figure S22. The CV results of the electrochemical experiments.

Comments 3: In Fig. 1g, the interlayer distance of HsGDY is difficult to recognize at the current scale.

Response: We are sorry that the low definition of the referenced image causes some of the content to be blurred, especially in Figure 1g. The ideal single-layer HsGDY is normally scarce

as a result of π - π interaction and van der Waals force. Based on your suggestions, a clearer HRTEM image of as-prepared HsGDY is displayed in the revised manuscript as shown in Figure 1g. It can be clearly seen that the interlayer distance of sample is about 0.416 nm, which is the characteristic value for the multilayer HsGDY. Thanks for your kind suggestions.

Revisions: In the revised manuscript, on page 12, Figure 1g is modified. Now it has higher definition to clearly show that the sample has the interlayer distance of 0.416 nm.

Figure 1g. The HRETEM image of fresh HsGDY.

Comments 4: *In Fig. 2e, what was the test time for SO₂ impact experiments? What about the effect of other typical flue gas components, e.g., H₂O, NO_x.*

Response: For each test, the gas stream was first switched to bypass and the inlet gas was sampled to acquire stable Hg⁰ concentrations (around 30 min). Thereafter, the Hg⁰ containing gas flow was passed through the reactor for Hg⁰ adsorption tests under the presence of SO₂. In Figure 2e, the test time for SO₂ impact experiments is 90 min. As it is shown, the Hg⁰ removal efficiency of HsGDY maintains at around 97.5% under 600 to 2400 ppm SO₂ conditions, suggesting its excellent anti-SO₂ ability. Considering that NO and H₂O are also typical components in real flue gas and might influence Hg⁰ removal, we have examined the Hg⁰ adsorption performance of HsGDY in the gas stream containing different contents of NO and H₂O. The results have been added in Figure S9 in the Supplementary Information of the revised manuscript. It can be seen that NO slightly promotes (at least not inhibitory) the Hg⁰ adsorption

over HsGDY. Some existing studies also indicate the slight promotive effect of NO on Hg⁰ adsorption. [*Sci. Total Environ.* 2019, 652: 651-659; *J. Environ. Chem. Eng.* 2021, 9: 105993] Figure S9b exhibits the Hg⁰ adsorption performance over HsGDY under different H₂O concentrations. When 1 %vol. H₂O is introduced into the N₂ + Hg⁰ atmosphere, the Hg⁰ removal efficiency of HsGDY decreases from around 97.4% to 90.4%. As the H₂O concentration is further increased to 3%vol., a decrease of Hg⁰ removal efficiency to 87.7% is observed. Above results indicate that H₂O slightly inhibits Hg⁰ removal over HsGDY, which is consistent with the literatures. The inhibitive effect can be attributed to the competitive adsorption between Hg⁰ and H₂O over the surface active sites. [*Chem. Eng. J.* 2017, 324: 279-286; *Chem. Eng. J.* 2021, 421: 127745] Although H₂O occupies part of the active sites of HsGDY, there are still more available active sites for Hg⁰ adsorption due to the ordered porous structure and highly accessible adsorption sites of HsGDY. The Hg⁰ removal activity of HsGDY in the simulated flue gas has been also examined for its potential application as shown in Figure 3d. The simulated flue gas containing 340 µg/m³, 600 ppm SO₂, 600 ppm NO, 5% vol. O₂, 3% vol. H₂O and balanced N₂ passed through the filter mediums and the outlet Hg⁰ concentration was detected. It is shown that more than 90% of Hg⁰ can be removed by the HsGDY modified filter mediums under simulated flue gas conditions. For the mercury vapor capture in the production workshops (H₂O content in ambient air less than 1% vol.), the Hg⁰ adsorption over HsGDY will be naturally less influenced by H₂O presence. Of course, there is a difference between the simulated gas atmosphere and real flue gas. It deserves a further study. Thank you for your good comments.

Revisions: We have performed the experiments of the effects of NO and H₂O on Hg⁰ adsorption over HsGDY and the results are displayed in Figure S9 in the Supplementary Information of the revised manuscript. The description of the effects of NO and H₂O on Hg⁰ adsorption over HsGDY in the text (page 14) are modified as follows: “Considering that SO₂, NO and H₂O are typical components in real flue gas and might influence Hg⁰ removal, we further examined the Hg⁰ adsorption performance of HsGDY in the gas stream containing different contents of SO₂/NO/H₂O. As shown in Supplementary Fig. 9, NO slightly promotes (at least not inhibitory) the Hg⁰ adsorption over HsGDY. H₂O presence slightly decreases the Hg⁰ removal efficiency (from 97.4% to 87.7% under 3 %vol. H₂O) over HsGDY, which can be

attributed to the competitive adsorption between Hg^0 and H_2O over the adsorption sites.”

Figure S9. The effects of NO and H_2O on Hg^0 adsorption over HsGDY.

Comments 5: In Fig. S12 and Fig 5b, the binding energy at 104 eV cannot be assigned to Hg^{2+} . The Hg^{2+} has spin-orbit doublet peaks, Hg 4f_{7/2} and Hg 4f_{5/2} respectively.

Response: Thanks for your professional comments on our manuscript. Careful deliberation of this issue has given us several ideas. Yes, it is not suitable to use “ Hg^{2+} ” to describe the state of adsorbed Hg. The description of “ Hg^{2+} ” has been replaced by “Hg with partially oxidized state” or other similar expressions in the revised manuscript. As shown in Figure 5b, compared to the Hg-laden HsGDY, the characteristic peak of 104 eV disappears in the thermally regenerated HsGDY. This observation indicates that Hg^0 adsorption onto HsGDY contributes to the appearance of the characteristic peak at 104 eV. Acting on your recommendation, we have studied many references associated with mercury removal. Shi et al. studied the Hg^0 adsorption over chlorine-based hierarchically porous biochar with CaCO_3 as template. [*Chem. Eng. J.* 2021, 406: 126828] After the mercury adsorption testing, the new peak appeared at 103.7-104.1 eV, which could belong to Hg^{2+} . Ma et al. studied the Hg^0 adsorption over modified layered ITQ-2 zeolites. [*J. Hazard. Mater.* 2022, 423: 127118] After Hg^0 adsorption, the Hg 4f XPS spectrum was characterized by two individual peaks attributed to HgO at 103 eV and HgCl_2 at 101.5 eV. It is noted that the single characteristic peak at 104 eV in this work is different from the spin-orbit doublet peaks of pure HgO, HgS and HgCl_2 (Hg 4f_{5/2} at 104-105 eV and Hg 4f_{7/2} at 100-101 eV; NIST X-ray Photoelectron Spectroscopy Database (SRD 20), Version 5.0 at <https://srdata.nist.gov/xps>). It is attributed to that the existence form and chemical state of the

formed Hg species over HsGDY surface is different from abovementioned mercury compounds with stable structure. Anyway, the characteristic peak of 104 eV suggests that gas-phase Hg atoms are successfully adsorbed onto the HsGDY surface. We consider that the adsorbed Hg atoms mainly exist in the form of partially oxidized state, which could be also verified by the results of the electrochemical experiments (Figure S22). The adsorbed Hg will lose some electrons which is transferred to sp-hybridized carbon due to the interaction of Hg with HsGDY. This effective interfacial electron transfer, namely strong electron-metal-support interaction, greatly boosts the immobilization of gas-phase Hg⁰ on HsGDY. Based on your suggestions, to make it more scientific, we have assigned the characteristic peak at 104 eV to the adsorbed Hg with partially oxidized state in the revised manuscript. In addition to XPS, there are other experimental results such as Hg-TPD and DFT calculations can support the interaction between Hg⁰ and HsGDY with the occurrence of charge transfer.

Revisions: As you said, it is not suitable to use “Hg²⁺” to describe the state of adsorbed Hg. The description of “Hg²⁺” has been replaced by “Hg with partially oxidized state” or other similar expressions in the revised manuscript.

(1) “The oxidation process is driven by the surface charge heterogeneity of HsGDY which can itself induce the adsorbed Hg atoms to lose electrons and present an oxidized state” has been replaced by “The oxidation process is driven by the surface charge heterogeneity of HsGDY which can itself induce the adsorbed Hg atoms to lose electrons and present a partially oxidized state” .

(2) “.....further reveals that gas-phase Hg atoms are successfully adsorbed on the HsGDY surface, mainly present in the form of high valent state Hg²⁺” has been modified as “further reveals that gas-phase Hg atoms are successfully adsorbed on the HsGDY surface, mainly present in the form of partially oxidized state” in the revised manuscript (page 20).

Comments 6: *The authors used thermal treatment for HsGDY recycle, while used acid leaching method for mercury recovery. These two methods obviously cannot be used at the same time, please provide an explanation.*

Response: Thanks for your comments. We really agree with you that the two methods (thermal regeneration and acid leaching) cannot be employed at the same time. These two methods may

have their own applicable scenarios. For the thermal regeneration, this method may be employed in the mercury recycle from flue gas (e.g. the coal fired power plants and smelting plants). The spent sorbent can be thermally regenerated by utilizing the waste heat in flue gas or heat-exchange facility to reduce the operating cost. As shown in Figure 2f, the Hg⁰ adsorption efficiency of HsGDY does not decline significantly in the 8 cycles of Hg⁰ capture and thermal regeneration, which facilitates its reusability. After thermal treatment, the evaporated mercury can be condensed and recycled as a by-product. For the acid leaching, this method may be employed in the mercury recycle from mercury-related production workshop. The mercury can be desorbed from the mercury-laden HsGDY by acid washing. Afterwards, the leached mercury in the solution can be effectively precipitated by Na₂S after adjusting pH and the formed HgS can be recovered and processed into raw materials for industrial production again. Therefore, suitable and cost-effective regeneration methods can be adopted according to the actual situation.

Comments 7: *In Line 321-323, the experiments of HgS formation should be added, and the XRD pattern of the resultant precipitates should also be provided.*

Response: Thanks for your professional comments. Acting on your recommendation, we have performed the experiments of HgS precipitation. HsGDY is treated by Hg-containing gas flow more than 16 h to ensure enough Hg on HsGDY to form visible HgS precipitate. Briefly, the mercury-laden HsGDY is firstly desorbed by acid leaching. The details about mercury desorption by acid washing has been displayed in the Methods (page 31) of the revised manuscript. Then the pH of the obtained solution is adjusted to 7-8 by NaOH. Afterwards, Na₂S dilute solution is dropwisely added into the aforementioned solution under continuously stirring at room temperature. Finally, the formed precipitate is separated by filtration and washed several times by deionized water. It can be seen from Figure S12a that very small reddish brown or black particles are suspended in the solution after Na₂S addition. This observation can demonstrate that Hg⁰ has been adsorbed by HsGDY. The adsorbed Hg on HsGDY can be recovered by acid leaching followed by precipitation. The accumulated resultant precipitate is further examined by XRD and the XRD pattern is displayed in Figure S12b in the Supplementary Information of the revised manuscript. As it can be seen from

Figure S12b, the diffraction peaks at 26.7° and 30.6° attributed to HgS (PDF 03-0396) is clearly observed in the pattern. Therefore, the leached mercury in the solution can be effectively precipitated by Na_2S and the formed HgS can be recovered and processed into raw materials for industrial production again.

Revisions: Acting on your recommendation, we have performed the experiments of HgS precipitation. The results of precipitation experiments and the XRD pattern of precipitate have been added in Figure S12 in the Supplementary Information of the revised manuscript.

Figure S12. The results of precipitation experiments and the XRD pattern of precipitate.

Comments 8: *In Fig. 4b, there are more than 3 bright atoms in the HAADF-STEM image, why do the authors consider the three circles indicated to be Hg atoms? The HAADF-STEM of raw HsGDY should be provided for comparison.*

Response: Thanks for your constructive suggestions. We really agree with you that there should be a comparison of the HAADF-STEM images between fresh HsGDY and spent HsGDY (Hg-laden) to clearly show that the isolated bright atoms are ascribed to the presence of Hg. Based on your advice, the fresh HsGDY is displayed in the revised manuscript as shown in Figure S16. It can be seen from Figure 4b that there are obviously isolated bright atoms anchored on the HsGDY support (relatively dark). Metal atoms will be brighter than the carbon atoms in a same HAADF-STEM image. In comparison with the spent HsGDY, no significant brightness difference is observed for the fresh HsGDY as shown in Figure S16. Therefore, the isolated bright atoms on spent HsGDY can be attributed to the presence of Hg.

Revisions: We have performed the HAADF-STEM characterization of the fresh HsGDY and

the results are displayed in Figure S16 in the Supplementary Information of the revised manuscript.

Figure S16. The HAADF-STEM image of fresh HsGDY.

Comments 9: *In Fig. 4c, the intensity of Hg element in the mapping image is too weak, which maybe the background error. The EDS spectra should be provided to further confirm the presence of Hg element.*

Response: Thanks for your valuable comments. The intensity of Hg element in the mapping image as shown in Figure 4c is weak. There are some reasons causes this observation. Briefly, the content of Hg on the spent HsGDY sample is relatively low due to the short adsorption time (90 min). The adsorbed Hg atoms are isolated. Above reasons causes the weak intensity of Hg element in the Hg mapping image of the elected area. According to your suggestions, we have performed the EDS characterization of the Hg-laden HsGDY. The EDS results have been displayed in the Figure S15 in the Supplementary Information of the revised manuscript. It can be found from Figure S15 that the absorbed Hg atoms are present on the surface of HsGDY. Besides, there are many other experimental results can strongly confirm the existence of Hg.

1) HAADF-STEM: The comparison of the HAADF-STEM images between fresh HsGDY (Figure S16) and spent HsGDY (Figure 4b) can clearly show that the isolated bright atoms are ascribed to the presence of Hg.

2) XPS: The comparison of the Hg 4f spectra between spent HsGDY and regenerated HsGDY (Figure 5b) can indicate the presence of Hg. The Hg 4f XPS analysis of Hg/HsGDY as shown in Figure 5b reveals that gas-phase Hg atoms are successfully adsorbed on the

HsGDY surface. The characteristic peak of 104 eV ascribed to partially oxidized Hg is not detected in the regenerated HsGDY sample.

3) Hg-TPD: The formed mercury compounds on HsGDY surface is examined by the temperature programmed desorption of Hg (Hg-TPD) analysis. As shown in Figure S8, a mercury desorption peak emerges at around 320 °C for HsGDY which can be ascribed to the decomposition/desorption of strongly bonded mercury species.

4) Raman: As displayed in Figure S13a, the Raman spectra of HsGDY after Hg⁰ adsorption (Hg/HsGDY) presents a shift in peak position and an increase in peak intensity compared to HsGDY, indicating the formation of coordination bonds between carbon atoms of conjugated diyne linkers (2021 cm⁻¹ and 2193 cm⁻¹) in HsGDY and Hg atoms. After thermal regeneration, the characteristic peaks in the regenerated HsGDY attributed to aromatic rings and acetylenic bonds are basically restored to the state of fresh sample.

5) Acid leaching and Na₂S precipitation: It can be seen from Figure S12 that very small reddish brown or black particles ascribed to HgS are suspended in the solution after acid leaching followed by Na₂S addition. This observation can demonstrate that Hg⁰ has been adsorbed onto HsGDY.

To summarize, all of above experimental results can strongly confirm the adsorption of Hg on to HsGDY. Thank you so much for providing these valuable comments.

Revisions: We have performed the EDS characterization of the Hg-laden HsGDY. The TEM-EDS results and SEM-EDS results have been displayed in the Figure S14 and Figure S15 in the Supplementary Information of the revised manuscript.

Figure S14. The TEM (a) and energy dispersive X-ray spectroscopy results (b) of Hg/HsGDY.

Figure S15. The SEM-EDS results of Hg/HsGDY.

Comments 10: *In line 411-414, the data or figures of OHM results are missing.*

Response: We are sorry that the OHM results are not displayed in the manuscript which confuses you. Based on your suggestions, the OHM results have been added in Figure S19 in the Supplementary Information of the revised manuscript. It can be seen that the desorbed mercury species during thermal regeneration are primarily present in the form of elemental mercury.

Revisions: The OHM results have been added in Figure S19 in the Supplementary Information of the revised manuscript.

Figure S19. The OHM results of the desorbed mercury species.

Comments 11: *In Fig. 4j and k, why use graphene for comparison of molecular dynamics simulations rather than GDY?*

Response: Thanks for your good comments. In this work, DFT calculations and molecular dynamic simulation are conducted to reveal the Hg atom adsorption behavior and the binding mechanism over the sp-hybridized carbon material (HsGDY). Graphene (GE) is a typical and commonly studied sp²-hybridized carbon material. Therefore, for comparison, the Hg atom adsorption over sp²-hybridized GE is investigated. According to your suggestions, we have performed the molecular dynamic simulation of Hg atom adsorption over graphdiyne (GDY) and the results have been displayed in Figure S17 in the Supplementary Information of the revised manuscript. It can be seen that the adsorption process of Hg atom from free state to stable adsorption state is accompanied with significant electron transfer for HsGDY. However, less electron transfer is observed for GDY and GE. Furthermore, the energy fluctuations of Hg atom adsorption over HsGDY, GDY and GE after reaching a relatively stable state are investigated and the results have been displayed in Figure S18. As shown in Figure S18, after Hg atom adsorbs onto HsGDY, it can reach a relatively stable state with smaller energy fluctuations. Hence, HsGDY can serve as an effective “trap” to anchor Hg atoms by strong electron-metal-support interaction, which is entirely different from GE and GDY.

Revisions: The molecular dynamic simulation results of Hg atom adsorption over GDY are added in Figure S17 in the Supplementary Information of the revised manuscript. The energy fluctuations of Hg atom adsorption over HsGDY, GDY and GE after reaching a relatively stable state are investigated and the results have been added in Figure S18 in the Supplementary Information of the revised manuscript. The description of Figure S18 has been added in the manuscript (page 21) as follows: “Additionally, the adsorption of Hg atom onto HsGDY can reach a relatively stable state with smaller energy fluctuations as shown in Supplementary Fig.18.”

Figure S17. The adsorption process of Hg atom from free state to stable adsorption state over GDY by AIMD simulation.

Figure S18. Energy fluctuations after Hg adsorption over HsGDY (a), GDY (b) and GE (c) by AIMD simulation.

Reviewer 3#

General Comments: *Hg capture research is important due to its direct harmful impact on people and environment. This research team developed a novel sorbent, HsGDY. The team did a great work in designing and performing tests and studying the associated reaction mechanisms. It is publishable in Nature Communications. However, a few questions need to be addressed.*

Response: Thanks for your positive comments and kind suggestions! We have improved the quality of the manuscript per your suggestions. Our point-to-point responses and detail revisions are listed below.

Comments 1: *Authors say that “It can be seen that the charge non-uniformity (σ) of HsGDY is obviously higher than that of GDY. This uneven charge distribution of HsGDY might induce electron redistribution between the HsGDY and adsorbed Hg atom, leading to a strong interaction between them”. What are the redistribution limits and the optimal redistribution? That are their differences? Also, how can the team control them with experiments?*

Response: We really appreciate your nice comments on our manuscript. According to the electron localization function (ELF) results as shown in Figure 1b and Figure S1b, we quantitatively compare the charge non-uniformity of GDY and HsGDY by borrowing the concept of dispersion degree and also considering the mercury binding sites (the center of triangular

hole for GDY and the side sites of acetylenic bond in plane for HsGDY). The detailed calculation method is displayed in the Methods of the revised manuscript (page 29). As shown in Figure S3, the charge non-uniformity (σ) of HsGDY is obviously higher than that of GDY. The uneven charge distribution characteristics of HsGDY might be attributed to that the large pore structure leads to lower local atomic density and resultantly uneven distribution of sp-hybridized and sp²-hybridized carbons in the local space. This uneven charge distribution of HsGDY might induce electron redistribution between the HsGDY and adsorbed Hg atom, leading to a strong interaction between them. This resultant electron transfer between HsGDY and adsorbed Hg atom can be confirmed by the experimental results such as Hg-TPD, XPS, etc. The DFT calculation results as shown in Figure 4d also indicate that significant electron redistribution occurs between sp-hybridized carbon and adsorbed Hg atom over HsGDY, in which electrons are accumulated on sp-hybridized carbon and depleted around the Hg atom.

For the electron redistribution, the redistribution limit may involve the energy conservation, charge conservation, and possible quantum mechanical rules during electron transfer. The optimal redistribution mentioned above refers to achieving the most stable or favorable state of the system through the redistribution of electrons and also satisfying the physical laws such as charge conservation and energy conservation, thereby achieving enhanced adsorption energy and improved charge transfer efficiency. The electronic structure changes of HsGDY after mercury adsorption were simulated by DFT, including the redistribution of electron density and charge transfer amount. Compared to sp²-hybridized carbon, sp-hybridized carbon in HsGDY may have a relatively higher redox potential due to its higher unsaturation and possible lower electron cloud density, which makes it easier to accept electrons in electrochemical environments.

Therefore, the surface charge heterogeneity of HsGDY which can itself induce the adsorbed Hg atoms to lose electrons and present a partially oxidized state. It should be emphasized that this is not the only factor that makes HsGDY better than GDY for Hg⁰ adsorption. The large hexagonal pore structure and “AB” stacking structure of HsGDY can facilitate the rapid diffusion of Hg atoms across the sorbent and make the mercury binding sites highly accessible. However, the diffusion of Hg atoms across GDY will be constrained due to its much smaller pore size and “ABC” stacking structure.

To summarize, such electron redistribution might be influenced by the adsorption configuration and local electron distribution of adsorbent and adsorbate. That is to say, Graphdiyne based materials with different chemical structures will have different surface charge distributions. Their adsorption capacities for Hg^0 will be resultantly different. In theory, if the pore structure and surface charge distribution can be adjusted, there might be an optimal chemical structure of graphdiyne based materials for Hg^0 adsorption. Fortunately, we have noticed a newly-published article which focuses on designing the sp-hybridized carbon content of graphdiyne by changing the precursors. [*Angew. Chem. Int. Ed* 2024, 63(23): e202401501] This work might give us some ideas for the subsequent research. In future, we can rationally design the graphdiyne material with optimized sp-hybridized carbon content and pore size, which might be more beneficial for Hg^0 adsorption. It is a very meaningful and valuable work to make in-depth and meticulous research of development of new graphdiyne material for the mercury as well as environmental pollution control. Considering above-mentioned aspects, a great deal of work will be done in the future. Thanks for your valuable comments.

Comments 2: *The adsorption-desorption proceeds through reversible electron transfer between Hg and HsGDY. Ideally, according to the principle, the regeneration peak in Figure 2f should be even. However, they are not. Why? Can the authors give a comprehensive explanation?*

Response: Thanks for your professional comments on our manuscript. We really agree with your opinion that the regeneration peaks in Figure 2f should be ideally even, according to the reversible electron transfer between Hg and HsGDY. However, they are not. Careful deliberation of this problem has given us several ideas. There may be several factors contributing to the slight change of regeneration peaks after several times of regeneration:

1) Firstly, the regeneration process might not completely refresh spent HsGDY. The residual mercury relatively hard to desorption may occupy the adsorption sites of HsGDY and influence the Hg^0 adsorption in the next cycle.

2) Secondly, the structure of HsGDY may change slightly due to the thermal regeneration. As shown in Figure 5c in the revised manuscript, the Raman characteristic peaks attributed to aromatic rings and acetylenic bonds for the regenerated HsGDY are still retained and basically restored to the state of fresh HsGDY. This slight structure change of HsGDY may affect the

adsorption of Hg^0 and result in the change of regeneration peaks.

3) Thirdly, some experimental errors may exist in the Hg adsorption-desorption experiments. For an example, the possible slight fluctuation of inlet Hg^0 concentration may influence the Hg^0 capture at a certain period of time. In this work, a relatively constant Hg^0 vapor is produced from a mercury permeation device with N_2 carrying. The principle of generation of Hg^0 vapor is that the evaporation rate of liquid mercury varies at different temperatures. Although the mercury permeation device has been calibrated, there still will be inevitable Hg concentration error due to N_2 carrying. Additionally, the Hg adsorption-desorption experiments are conducted continuously during 3000 min. The possible slight baseline drift of the online mercury detection device may contribute to the above-mentioned experimental phenomena.

So far, whether the change of regeneration peak is caused by the slight change of HsGDY structure, regeneration conditions and experimental errors during the regeneration process still needs to be further confirmed. Considering above-mentioned aspects, a great deal of work will be done in the future. In all, the Hg^0 adsorption efficiency of HsGDY does not decline significantly in the 8 cycles of Hg^0 capture and regeneration, indicating its excellent regeneration property. Thank you so much for providing these valuable comments.

Point-to-point response to the reviewers' comments

Reviewer 1#

General Comments: *The authors still need to do some revisions.*

Response: Thanks for your positive comments and kind suggestions! We have improved the quality of the manuscript per your suggestions. Our point-to-point responses and detail revisions are listed below.

Comments 1: *For Comment 1, the given responds are reasonable, but the authors need to provide corresponding descriptions or revisions in the manuscript, or, the readers may be confused.*

Response: Thanks for your suggestion. In general, a negative adsorption energy typically indicates an exothermic process, where the system releases energy upon adsorption (i.e., the adsorbed state is more stable than the free state). A positive adsorption energy does not automatically mean the adsorption potential is poor. From Table S2 to S5 in the revised manuscript, it can be seen that the calculated adsorption energy values of Hg^0 onto HsGDY, GDY, GE and CNT for the optimal adsorption configuration are 0.005 eV, -0.082 eV, 0.102 eV and -0.398 eV, respectively. GE has a positive adsorption energy for Hg^0 means that the adsorption of Hg^0 onto GE is relatively difficult. This is consistent with its poor Hg^0 adsorption performance as shown in Figure 2a and 2b in the revised manuscript. It is indicated that there is some correlation between the calculated Hg^0 adsorption energy and the Hg^0 adsorption performance. But adsorption energy for Hg^0 is not perfectly consistent with the Hg^0 adsorption performance. For an example, the adsorption energy of Hg^0 onto GDY is negative (-0.082 eV). However, the Hg^0 adsorption performance of GDY is evidently lower than HsGDY with Hg^0 adsorption energy of 0.005 eV. This is because adsorption is a complex process that involves many factors beyond just the calculated adsorption energy. The pore characteristics and surface chemical properties such as charge heterogeneity of the sorbent can all play roles in determining the adsorption performance of Hg^0 . As mentioned in the text, the capture of Hg^0 onto HsGDY is benefited from the large hexagonal pore structure and surface charge

heterogeneity of HsGDY. These unique properties of HsGDY can lead to a strong interaction of HsGDY with Hg atoms, even if the overall adsorption energy of Hg^0 onto HsGDY is not negative. In addition, the positive adsorption energies can be interpreted as activation energies for the adsorption process. This means that some energy input (e.g., heat) is required to drive the adsorption of Hg^0 onto HsGDY. This result is in accordance with the effect of temperature on and kinetic analysis of Hg^0 adsorption over HsGDY. An increase of temperature from 100 °C to 150 °C can slightly enhance the Hg^0 removal by HsGDY as shown in Figure 2d in the revised manuscript. In summary, the calculated adsorption energy of Hg^0 onto HsGDY (0.005 eV) may not indicate a spontaneous and exothermic adsorption process, it still provides insights into the relative stability and potential for Hg adsorption on the HsGDY surface. Acting on your recommendation, we have provided the corresponding descriptions about this to make it more scientific and rigorous. Thank you so much for providing these valuable comments.

Revisions: In the revised manuscript (page 7, line 14), the corresponding descriptions in the text are modified as follows: “It can be seen that the calculated adsorption energies of Hg^0 onto HsGDY, GDY, GE and CNT for the optimal adsorption configuration are 0.005 eV, -0.082 eV, 0.102 eV and -0.398 eV, respectively. The calculated Hg^0 adsorption energy is somewhat related to but not perfectly consistent with the Hg^0 adsorption performance. Adsorption is a complex process that involves many factors such as pore characteristics and surface chemical properties beyond just the calculated adsorption energy. For HsGDY, the Hg atom tends to be adsorbed at the side sites of acetylenic bond in plane with the adsorption energy of 0.005 eV. Overall, the adsorption energy of Hg^0 onto HsGDY is not negative but close to 0, indicating the relative stability and potential for Hg^0 adsorption over HsGDY with energy input.”

Comments 2: *For Comment 2, based on the responds, it seems that GDY can adsorb 3 Hg atoms. If that is the case, the maximum adsorption numbers of Hg atoms for GDY should be revised to 3.*

Response: We feel very sorry that there has been a writing error about the maximum adsorption numbers of Hg atoms for GDY in our response to your comment 2 in the previous Response to Decision Letter. The mistake is just existed in the previous response not in the text. According to the calculations, the theoretical adsorption capacity of Hg per unit mass for monolayer HsGDY is triple that of GDY (HsGDY/6Hg vs. GDY/2Hg). This indicates that HsGDY, likely a modified version of GDY, has a higher affinity or capacity for adsorbing Hg atoms. We are very thankful for your helpful comments on our manuscript.

Comments 3: *For Comment 3, NVT canonical ensemble represents a definite temperature. Is the AIMD simulation in Figure 5d at 573 K or 298.15 to 573 K? If it is at 573 K, why you marked 298.15 K in the figure? If not, how did this simulation carried out under NVT canonical ensemble with a definite temperature? Besides, there should be Spaces between numbers and letters. Please revise them in this figure and check all the manuscript.*

Response: Thank you for your careful review and valuable comments. The marking of 298.15 K in the figure is not right. The corrected temperature for this AIMD simulation is 573 K. We really agree with your opinions. Regarding the simulation under the NVT canonical ensemble with a definite temperature, in AIMD simulations, the temperature is controlled by adjusting the velocities of the particles in the system. In the NVT ensemble, the number of particles (N), the volume (V), and the temperature (T) are kept constant. To maintain a constant temperature, a thermostat algorithm is used to adjust the kinetic energy of the particles, thereby controlling their velocities. This ensures that the system remains at the specified temperature throughout the simulation. The AIMD simulation in Figure 5d is surely conducted at a definite temperature of 573 K, which corresponds to the NVT canonical ensemble representation. The marking of 298.15 K in the figure is not right. This might bring about much perplexity to the readers and should be corrected. The corrected temperature for this AIMD simulation is 573 K. Our original intention is to compare the adsorption and desorption process of Hg⁰ over HsGDY surface under two typical

temperatures (298.15K and 573 K) by AIMD simulation, however with an inappropriate expression in Figure 5d. We really apologize for the confusion caused by the incorrect temperature marking in the figure and have corrected it accordingly in the revised manuscript. We have also checked the entire manuscript for any similar mistakes and make the necessary corrections. In addition, the spaces have been added between numbers and letters in the revised manuscript and all the figures. We are very thankful for your helpful comments on our manuscript.

Revisions: The marking of 298.15K has been deleted from Figure 5d in the revised manuscript. Now Figure 5d has a right temperature marking of 573 K. Moreover, the spaces have been added between numbers and letters in the whole text. All the figures including Figure 1, Figure 4, Figure 5, Figure S18 and Figure S19 in the revised manuscript have been checked for it.

Figure 5d. The molecular dynamic simulation results of Hg desorption over HsGDY.

Reviewer 3#

General Comments: *The revision work was well done. It is publishable in its current form.*

Response: Thanks for your positive comments.

Reviewer 4#

General Comments: *This work proposes a graphdiyne material with accessible*

sp-hybridized carbons for mercury vapor capture, which has not been widely studied in carbon-based adsorbents, and this work is of great significance. The comments should be addressed before publication are listed as follows:

Response: Thanks for your positive comments and kind suggestions! We have improved the quality of the manuscript per your suggestions. Our point-to-point responses and detail revisions are listed below.

Comments 1: *The manuscript mentions that HsGDY has unique sp-hybridized carbon and large hexagonal pore structure. How do these characteristics compare with the existing mercury adsorption materials? Please explain in detail the fundamental differences and advantages between HsGDY and the existing technology in adsorption mechanism and performance. Please further explore the quantitative relationship between pore structure parameters (such as pore size, pore volume, pore distribution) and the adsorption performance of Hg⁰, and how to optimize the adsorption performance of materials by adjusting pore structure.*

Response: Thank you for your careful review and valuable comments. Careful deliberation of the issues has given us several ideas. The responses to your comments are listed as follows:

The disadvantages of typical existing mercury adsorption materials are as follows. Some good Hg⁰ sorbents with high adsorption capacities and SO₂ resistance such as sulfur-based materials have been reported in literatures. [Chem. Eng. J. 2021, 411: 128608; Environ. Sci. Technol. 2022, 56: 13664-13674] Selenium-based material also exhibits excellent Hg⁰ adsorption capacity due to its strong affinity to mercury. [J. Environ. Sci. 2025, 148: 420-436; Chem. Eng. J. 2023, 453: 139946] These reported typical sorbents have promising prospects in different industrial application scenarios. However, the deactivated mercury sorbent maybe considered as a hazardous waste. In addition to its Hg⁰ removal performance, the stability, toxicity and secondary pollution risk of the sorbents are also worthy of attention. Elemental sulfur and selenium species active for Hg⁰ adsorption are thermally unstable (low melting point, prone to sublimation). [Environ. Sci. Technol. 2013, 47: 10056-10062] The possible leakage and loss of introduced sulfur and selenium species during sorbent application

and thermal regeneration would cause the performance decline as well as secondary pollution. The active sulfur and selenium species of the sorbent needs to be replenished by H₂S treatment, re-selenization, etc., to maintain the sorbent activity. [*Environ. Sci. Technol.* 2018, 52(17): 10003-10010; *J. Hazard. Mater.* 2021, 406: 124744; *Chem. Eng. J.* 2020, 394: 125022] Sulfur and selenium species have certain toxicity and are important sources of groundwater pollution. [*Environ. Pollut.* 2022, 299: 118858] In addition to mercury removal from flue gas, populations with occupational exposure to mercury also needs to be paid attention to. Personal protection such as wearing gas mask is of great significance to meet the occupational health requirements. Sulfur and selenium-based materials are not applicable to the filter medium of gas mask due to their toxicities. Although sulfur and selenium-based materials show good abilities on Hg⁰ removal, some existing problems abovementioned still needs to be seriously considered. The porous structure and environmental friendliness of carbon material makes it a promising candidate for mercury vapor capture. Unfortunately, the weak interfacial bonding with mercury is still the choke point in efficient mercury capture by carbon material. The adsorption of Hg⁰ onto traditional carbon material is primarily due to the physical adsorption which is determined by the porous structure of carbon material. The interfacial bonding of Hg⁰ with carbon surface is generally weak. The electron transfer between adsorbed Hg⁰ and carbon material can enhance the Hg⁰ immobilization on carbon surface, which is hardly achieved by traditional sp²-hybridized material like graphene with uniform surface charge distribution. Therefore, most of today's carbon materials need to be modified by ligands with affinity towards mercury such as halogen, sulfur and selenium species to enhance their Hg⁰ adsorption performance. However, the possible leakage and loss of introduced species (e.g. toxic selenium) during sorbent application and regeneration would cause the performance decline as well as secondary pollution. The aforementioned problems necessitate the search for more efficient, sustainable and environmental-friendly carbon materials for mercury capture.

The advantages of HsGDY compared to the existing technologies in Hg⁰

adsorption performance are summarized as follows.

1) Excellent Hg⁰ capture ability: Developing efficient and sustainable mercury vapor removal technologies is an urgent task. In this work, the proposed carbon material HsGDY can serve as an effective “trap” to anchor Hg atoms by strong electron-metal-support interaction. The Hg⁰ adsorption performance of HsGDY is apparently superior to that of tested carbon materials (graphdiyne, carbon nanotube, graphene and activated carbon) and most of the other reported carbon sorbents such as the commercial Br-modified carbon sorbent DARCO® Hg-LH EXTRA.

2) Environmental friendliness: In our previous work, it is shown that HsGDY will not destroy the cell structure during hemoperfusion and has good biocompatibility. [*PNAS 2023, 120(16): e2221002120*] HsGDY is a novel carbon material and can achieve good Hg⁰ adsorption performance without additional chemical modification. HsGDY can be applied to the personal protective equipment (directly utilized in the filter layer of mask) to prevent the workers from mercury exposure. Moreover, As shown in Figure S21 in the revised manuscript, the regenerated HsGDY (thermal treatment at 400 °C under N₂ stream) still exhibits a porous structure and its area ratio of C-C (sp²) to C-C (sp) is similar to that of fresh sample, indicating the structural stability of the carbon skeleton of HsGDY. This means that there will be less leakage and loss of sorbent species during its application and thermal regeneration. Therefore, it brings about less secondary pollution risk when HsGDY is applied for mercury vapor removal. We believe that the potential environmental impact of the sorbent needs to be carefully considered in practical application scenarios.

3) Excellent regeneration performance: As shown in the manuscript, the surface charge heterogeneity of HsGDY can induce the adsorbed Hg atoms to lose electrons and present a partially oxidized state. This electron transfer between Hg and HsGDY is reversible. Hg-laden HsGDY can be restored to the original state after Hg desorption as shown in Figure 5. The adsorbed Hg atoms that have lost electrons to HsGDY can recapture the electrons and are released to the gas stream in the form of elemental mercury by heat treatment, which can be also verified by the molecular

dynamic simulation results of Hg desorption over HsGDY at 300 °C as shown in Figure 5d. Such reversible electron transfer between Hg and HsGDY and structural stability of HsGDY can give it excellent thermal regeneration performance. The regeneration performance of sorbent is essential to its real application. As shown in Figure 2f, the Hg⁰ adsorption efficiency of HsGDY does not decline significantly in the 8 cycles of Hg⁰ capture and regeneration. It is indicated that HsGDY has excellent regeneration property and application potential for mercury vapor capture. In comparison, the inevitable loss of sulfur species during thermal regeneration would eventually cause the performance decline of sulfur-based sorbent. The active sulfur species of the sorbent needs to be replenished by sulfide impregnation or H₂S treatment, etc. Although the adsorption performance of HsGDY is not as good as sulfur-based material, its good regeneration performance can compensate for it.

The fundamental differences between HsGDY and the existing technologies in Hg⁰ adsorption mechanism are as follows. The Hg⁰ adsorption onto the traditional carbon material is primarily due to the physical adsorption determined by the porous structure of carbon material. The electron transfer between adsorbed Hg⁰ and carbon material itself is scarcely observed. The large hexagonal pore structure aside, HsGDY with surface charge heterogeneity has accessible sp-hybridized carbons which can itself realize a unique in-situ adsorption-coupled-oxidation of gas-phase elemental mercury, which enables its ultrahigh efficient mercury vapor capture. The DFT calculations as shown in Figure 4 in the revised manuscript indicates that significant electron redistribution occurs between sp-hybridized carbon and adsorbed Hg atom, in which electrons are accumulated on sp-hybridized carbon and depleted around the Hg atom. However, no obvious electron transfer between Hg atom and sp²-hybridized carbon in graphene can be detected for the adsorption configuration of Hg/graphene. There are also experimental results such as XPS (Figure 5b), Hg-TPD (Figure S9) and electrochemical experiments (Figure S23) to support the strong electron-metal (Hg) -support (HsGDY) interaction. To summarize, the surface charge heterogeneity of HsGDY with accessible sp-hybridized carbons can itself induce the adsorbed Hg atoms to lose electrons and present a partially oxidized state, which leads

to the strong bonding of mercury with HsGDY surface. This has not been previously accessed in another carbon material.

HsGDY has unique sp-hybridized carbons and large hexagonal pore structure. **The comparison of HsGDY with such characteristics and the existing mercury adsorption materials is as follows.** The traditional sp²-hybridized carbon (C=C) materials such as graphene (GE) show the common characteristics of uniform surface charge distribution, which might have weak charge transfer interaction with mercury atom. Graphdiyne-like material has a huge two-dimensional planar hyperconjugated structure, which can be regarded as an “electron sponge”. Due to its large highly conjugated structure, it has become a good electron acceptor [*ACS Nano* 2013, 7(2): 1504-1512; *Small* 2012, 8(2):265-271]. The alkyne bond (C≡C) can be seen as an alkene bond (C=C) with two hydrogen atoms removed, which means that when it exists in the form of an alkyne bond, it loses some of its negative charge (net electron density), so the alkyne bond exhibits electron deficient properties. [*Acta Phys. -Chim. Sin.* 2018, 34 (9): 1048-1060] HsGDY has lower atom density and more sp-hybridized carbons in the pores, which might lead to higher Hg⁰ adsorption capacity. The presence of sp-hybridized carbon in HsGDY endows it with electron deficient properties as well as surface charge heterogeneity. This contributes to the electron transfer between adsorbed Hg and HsGDY. As for pore structure, the ordered and large pore structure is theoretically beneficial to the Hg⁰ diffusion across the sorbent. For an example, although GDY is also a sp-hybridized carbon material similar to HsGDY, it is defeated by HsGDY for mercury capture due to its constrained mass transfer (Figure S1e) and relatively lower surface charge heterogeneity.

Acting on your recommendation, we have further explored the quantitative relationship between pore structure parameters such as pore size and pore volume and the Hg⁰ adsorption performance. The BJH pore size distribution curves of GE, CNT, GDY, HsGDY and AC are displayed in Figure R1. The laws of variation of Hg⁰ adsorption capacity with average pore size and pore volume are exhibited in Figure S6. It can be seen that there is some correlation between the Hg⁰ adsorption performance and the pore characteristics. In the range of pore volume from

0 to 0.4 cm³/g, the Hg⁰ adsorption performance can increase with the increase of pore volume. But the pore volume or pore size is not perfectly consistent with the Hg⁰ adsorption performance. An inverted U curve can be observed. For an example, the pore volume of HsGDY is 0.3758 cm³/g, which is obviously lower than that of CNT and AC. However, the Hg⁰ adsorption performance of HsGDY is evidently better than CNT and AC. This is because adsorption is a complex process that involves many factors beyond just the pore volume and pore size. It should be emphasized that the pore characteristics of sorbent is not the only factor that affects Hg⁰ adsorption. The pore characteristics and surface chemical properties such as charge heterogeneity of the sorbent can all play roles in determining the adsorption performance of Hg⁰. As mentioned in the text, the capture of Hg⁰ onto HsGDY is benefited from the large hexagonal pore structure and surface charge heterogeneity of HsGDY. These unique properties of HsGDY can lead to a strong interaction of HsGDY with Hg atoms, even if the BET surface area as well as pore volume of HsGDY is not the highest.

Finally, we really agree with your opinion that the pore structure may be adjusted to achieve the optimized Hg⁰ adsorption performance by graphdiyne based materials. As aforementioned, the electron redistribution might be influenced by the adsorption configuration and local electron distribution of adsorbent and adsorbate. That is to say, graphdiyne based materials with different chemical structures will have different surface charge distributions. Their adsorption capacities for Hg⁰ will be resultantly different. In theory, if the pore structure and surface charge distribution can be adjusted, there might be an optimal chemical structure of graphdiyne based materials for Hg⁰ adsorption. Fortunately, we have noticed a newly-published article which focuses on designing the sp-hybridized carbon content of graphdiyne by changing the precursors. [Angew. Chem. Int. Ed 2024, 63(23): e202401501] This work might give us some ideas for the subsequent research. In future, we can rationally design the graphdiyne material with optimized sp-hybridized carbon content as well as pore size, which might be more beneficial for Hg⁰ adsorption. It is a very meaningful and valuable work to make in-depth and meticulous research of development of new graphdiyne material for the mercury as

well as environmental pollution control. Considering above-mentioned aspects, a great deal of work will be done in the future.

Revisions:

1) In the revised manuscript (page 23, line 13), a description about the advantages of HsGDY and the comparison of HsGDY with other existing mercury adsorption materials have been given. The related sentences are as follows: “The Hg⁰ adsorption onto traditional carbon material is primarily due to the physical adsorption determined by the porous structure of carbon material. The interfacial bonding of Hg⁰ with carbon surface is generally weak. The electron transfer between adsorbed Hg⁰ and carbon material can enhance the Hg⁰ immobilization on carbon surface, which is hardly achieved by traditional sp²-hybridized graphene with uniform surface charge distribution. Although GDY is also a sp-hybridized carbon material similar to HsGDY, it is defeated by HsGDY for mercury capture due to its constrained mass transfer and relatively lower surface charge heterogeneity. Moreover, different from toxic sulfur and selenium compounds, the environmental-friendly HsGDY can be applied to the personal protective equipment, greatly broadening its application fields. Overall, a unique in-situ Hg⁰ adsorption-coupled-oxidation process can be achieved by HsGDY, which enables its ultrahigh efficient mercury vapor capture and thus effectively solves the bottleneck problem of the weak interfacial bonding between mercury and traditional carbon sorbents.”

2) We have given a quantitative analysis of the relationship between pore structure parameters such as pore size and pore volume and the Hg⁰ adsorption performance. The laws of variation of Hg⁰ adsorption capacity with average pore size and pore volume are exhibited in Figure S6.

Figure R1. The BJH pore size distribution curves of GE, CNT, GDY, HsGDY and AC.

Figure S6. The variation of Hg^0 adsorption capacity with average pore volume (a) and pore width (b).

Comments 2: *In this paper, it is mentioned that HsGDY achieves high-efficiency adsorption of Hg^0 through sp -hybridized carbon. Please provide more detailed theoretical calculation or simulation data, such as density functional theory (DFT) calculation to reveal the molecular mechanism of the interaction between Hg^0 and HsGDY, including electron density distribution, orbital hybridization and possible chemical bond formation.*

Response: Thanks for your professional comments on our manuscript. DFT calculations have been conducted to reveal the Hg atom adsorption behavior and the binding mechanism over sp -hybridized carbon (HsGDY). The detailed theoretical calculation or simulation data such as DFT calculation including electron density distribution and orbital hybridization has been provided in Figure 4 in the revised manuscript. The charge density difference in Figure 4d indicates that significant electron redistribution occurs between sp -hybridized carbon and adsorbed Hg atom, in which electrons are accumulated on sp -hybridized carbon and depleted around the Hg atom. In detail, the Hg atom loses $0.27 e^-$ which is transferred to sp -hybridized carbon, thus existing in a partially oxidized state. The two-dimensional projection of

differential charge density contours of Hg/HsGDY results also clearly illustrate this electron transfer behavior between Hg atom and sp-hybridized carbon (Figure 4e). Additionally, the calculated partial density of states (PDOS) exhibits an orbital overlap between H1s, Hg5d, Hg6s and C2p below the Fermi energy level (~ 0 eV) for the adsorption configuration of Hg/HsGDY as shown in Figure 4f. Furthermore, we have employed the Ab initio molecular dynamics (AIMD) simulation to reveal the molecular mechanism of the interaction between Hg⁰ and HsGDY. The AIMD simulation results as displayed in Figure 4j clearly show the adsorption process of Hg atom from free state to stable adsorption state, which is accompanied with significant electron transfer between the adsorbed Hg atom and HsGDY. Thank you so much for providing these valuable comments.

Figure 4. The Hg^0 adsorption mechanism over HsGDY. **a** The high-resolution transmission electron microscopy (HR-TEM) image of Hg/HsGDY; **b** The HAADF-STEM image of Hg/HsGDY; **c** The HAADF-STEM image and corresponding element mapping images of C (red) and Hg (green); **d** The charge density difference of Hg/HsGDY (depletion and accumulation spaces are revealed in blue and yellow, respectively); **e** The two-dimensional projection of differential charge density contours of Hg/HsGDY; **f** The PDOS comparison for Hg 5d, Hg 6s, C 2p and H 1s orbitals within Hg/HsGDY; **g** The charge density difference of Hg/GE; **h** The two-dimensional projection of differential charge density contours of Hg/GE; **i** The PDOS comparison for Hg 5d, Hg 6s and C 2p orbitals within Hg/GE; **j** The molecular dynamic

simulation results of Hg adsorption over HsGDY; **k** The molecular dynamic simulation results of Hg adsorption over GE.

Comments 3: *In the multi-component gas environment, the selectivity of adsorption materials is very important. Has the author studied the selective adsorption behavior of Hg⁰ and other coexisting gases (such as SO₂, NO_x, CO, etc.) by HsGDY? Is there any data to support the preferential adsorption of Hg⁰ by HsGDY in complex gas mixture?*

Response: We appreciate your nice comments on our manuscript. We really agree with your opinion that the selectivity of adsorption material is very important when it is applied in the multi-component gas environment. Therefore, per your suggestion, the selectively adsorption of Hg⁰ over HsGDY in the presence of multiple gas components such as SO₂, NO, H₂O and CO has been added in the revised manuscript. Moreover, we have employed density functional theory (DFT) calculations to reveal the preferential adsorption of Hg⁰ over HsGDY. As shown in Figure 2e, the Hg⁰ removal efficiency of HsGDY maintains at around 97.5% under 600 to 2400 ppm SO₂ conditions. As shown in Figure S11a, the adsorption energies of the SO₂ molecule at different adsorption sites over HsGDY are all nearly 3.5 eV, which is obviously higher than that of Hg atom (~0 eV). Above exciting results manifest that Hg⁰ is preferentially adsorbed on HsGDY, leading to its excellent anti-SO₂ ability. As shown in Figure S10a, NO slightly promotes (at least not inhibitory) the Hg⁰ adsorption over HsGDY. As shown in Figure S11b in the revised manuscript, the adsorption energies of the NO molecule at different adsorption sites over HsGDY are all approximately 3.2 eV, which is apparently larger than that of Hg atom. That is to say, Hg⁰ is preferentially adsorbed on HsGDY and the Hg⁰ adsorption will be hardly influenced by NO presence. For the effect of H₂O on Hg⁰ removal over HsGDY as shown in Figure S10b, it can be seen that H₂O presence slightly decreases the Hg⁰ removal efficiency (from 97.4% to 87.7% under 3 %vol. H₂O) over HsGDY, which can be attributed to the competitive adsorption between Hg⁰ and H₂O over the adsorption

sites. As shown in Figure S11c in the revised manuscript, the adsorption energies of the H₂O molecule at different adsorption sites over HsGDY are in the range of 0.01-0.05 eV, which are higher than that of Hg atom. As shown in Figure 10c, the presence of CO just slightly decreases the Hg⁰ removal efficiency over HsGDY. As shown in Figure S11d, the adsorption energy of the CO molecule over HsGDY for the optimal adsorption configuration is around 0.01 eV, which is slightly higher than that of Hg atom. Although H₂O and CO slightly inhibit the Hg⁰ removal over HsGDY, HsGDY can still achieve more than 85% Hg⁰ removal, suggestive of its good selective adsorption capability for Hg⁰. Additionally, as shown in Figure 3d in the revised manuscript, it can be seen that HsGDY can achieve ~96% Hg⁰ removal under multi-component gas atmosphere containing NO, SO₂, H₂O, etc. To summarize, according to the initial judgement, HsGDY has satisfactory capability for the selective adsorption of Hg⁰ in the multi-component gas environment based on the experimental as well as DFT calculation results aforementioned. Certainly, there is a great difference between the simulated gas atmosphere and real flue gas conditions. It deserves a further study to systematically investigate the performance of HsGDY under the real and complicated gas conditions. The related investigation is currently underway. Thank you so much for providing these professional comments.

Revisions: We have further calculated the adsorption energies of SO₂/NO/H₂O/CO on HsGDY. The possible adsorption configurations and adsorption energies of SO₂/NO/H₂O/CO on HsGDY have been displayed in Figure S11a, Figure S11b, Figure S11c and Figure S11d, respectively. We have further examined the effects of gas components on Hg⁰ removal over HsGDY. The effects of SO₂, NO, H₂O and CO on Hg⁰ removal over HsGDY have been displayed in Figure 2e, Figure S10a, Figure S10b and Figure S10c, respectively.

Figure 2e. The effect of SO₂ on Hg⁰ adsorption over HsGDY.

Figure S10. The effects of NO (a), H₂O (b) and CO (c) on Hg⁰ adsorption over HsGDY.

Figure S11. The possible adsorption configurations and adsorption energies of SO_2

(a), NO (b), H_2O (c) and CO (d) onto HsGDY.

Reviewer #2 (Remarks to the Author) - First round of review

This work developed a graphdiyne material with accessible sp-hybridized carbons for mercury vapor capture. However, the developed material lags behind existing reported mercury adsorption materials and the proposed mechanisms lack innovation and are not well explored. Thus, I do not recommend its publication in Nature Communications.

The related comments are as follows:

- 1) What are the advantages of carbon-based materials compared to other materials? According to the literature, the adsorption capacities and SO₂ resistance of sulfur-based materials are much higher than those of carbon-based materials?
- 2) Generally, the removal of Hg⁰ by solid materials contains adsorption and oxidation process. Thus, the proposed “adsorption-coupled-oxidation” mechanism for Hg⁰ removal is not innovative. Moreover, the oxidation mechanism in this work is not well discussed.
- 3) In Fig. 1g, the interlayer distance of HsGDY is difficult to recognize at the current scale.
- 4) In Fig. 2e, what was the test time for SO₂ impact experiments? What about the effect of other typical flue gas components, e.g., H₂O, NO_x.
- 5) In Fig. S12 and Fig 5b, the binding energy at 104 eV cannot be assigned to Hg²⁺. The Hg²⁺ has spin-orbit doublet peaks, Hg 4f_{7/2} and Hg 4f_{5/2} respectively.
- 6) The authors used thermal treatment for HsGDY recycle, while used acid leaching method for mercury recovery. These two methods obviously cannot be used at the same time, please provide an explanation.
- 7) In Line 321-323, the experiments of HgS formation should be added, and the XRD pattern of the resultant precipitates should also be provided.
- 8) In Fig. 4b, there are more than 3 bright atoms in the HAADF-STEM image, why do the authors consider the three circles indicated to be Hg atoms? The HAADF-STEM of raw HsGDY should be provided for comparison.
- 9) In Fig. 4c, the intensity of Hg element in the mapping image is too weak, which may be the background error. The EDS spectra should be provided to further confirm the presence of Hg element.
- 10) In line 411-414, the data or figures of OHM results are missing.
- 11) In Fig. 4j and k, why use graphene for comparison of molecular dynamics simulations rather than GDY?